# Increased exosome secretion in neurons aging in vitro by NPC1-mediated endosomal cholesterol buildup

Francesc X Guix[1], Ana Marrero Capitán[1], Álvaro Casadomé-Perales[1], Irene Palomares-Pérez[1], Inés López del Castillo[1], Verónica Miguel[2], Leigh Goedeke[4,5], Mauricio G Martín[6], Santiago Lamas[2], Héctor Peinado[3], Carlos Fernández-Hernando[4,5], Carlos G Dotti[1]

As neurons age, they show a decrease in their ability to degrade proteins and membranes. Because undegraded material is a source of toxic products, defects in degradation are associated with reduced cell function and survival. However, there are very few dead neurons in the aging brain, suggesting the action of compensatory mechanisms. We show in this work that ageing neurons in culture show large multivesicular bodies (MVBs) filled with intralumenal vesicles (ILVs) and secrete more small extracellular vesicles than younger neurons. We also show that the high number of ILVs is the consequence of the accumulation of cholesterol in MVBs, which in turn is due to decreased levels of the cholesterol extruding protein NPC1. NPC1 down-regulation is the consequence of a combination of upregulation of the NPC1 repressor microRNA 33, and increased degradation, due to Akt-mTOR targeting of NPC1 to the phagosome. Although releasing more exosomes can be beneficial to old neurons, other cells, neighbouring and distant, can be negatively affected by the waste material they contain.

## Introduction

Among the many changes that occur with age, one of the most feared is the loss of cognitive abilities. A number of studies on age-associated cognitive alterations have shown that cognitive deficits of the old are not the consequence of neuronal loss (Reviewed by Bishop et al [2010]) but rather of a panoply of biochemical and molecular alterations that, when added together, lead to the functional deficits characteristics of the old age. Among the derangements in molecular processes, those related to proteostasis are gaining momentum (Hipp et al, 2019). Deficits in proteostasis encompasses defects in the process responsible for the biogenesis, folding, trafficking and degradation of proteins and membranes, resulting in the accumulation of misfolded and/or aggregated proteins in lysosomes and phagosomes. The defects in proteostasis with age are of multifactorial origin, including changes in the expression of genes related to folding and degradation, alterations in endocytic sorting and accumulation of oxidized proteins, inherently resistant to degradation (Mattson & Arumugam, 2018; Lévy et al, 2019). In turn, dysfunctional proteostasis may impair cellular function through several mechanisms, such as increased generation and release to the cytosol of oxidative products that will affect, among others, key synaptic structural and functional proteins, or by triggering changes which impede proper organelle interactions (e.g., lysosome-ER).

Even though numerous studies have shown that proteostasis defects do not only lead to neuronal dysfunction but also to their death (Mattson & Arumugam, 2018), the clinical cognitive correlate is much less dramatic, as most cognitive deficits in physiological aging are not conspicuous. How are then age-associated proteostasis defects neutralized so that its deleterious consequences do not greatly affect our usual abilities? Intuitively, one mechanism that neurons could put to work is by increasing the removal of potentially toxic and undegraded material by increasing the formation and release of extracellular vesicles (EVs). In fact, in conditions where the mechanisms for the degradation of cellular content are impaired, such as in neurodegenerative diseases, the release of EVs was able to restore cell homeostasis (Mathews & Levy, 2019).

EVs consist of lipid bilayers encapsulating a part of the cytosol and are generated by every cell type (Raposo & Stoorvogel, 2013). A recent classification based on size divided them into small EVs

[1]Molecular Neuropathology Unit, Physiological and Pathological Processes Program, Centro de Biología Molecular Severo Ochoa, Consejo Superior de Investigaciones Científicas (CSIC)/Universidad Autónoma de Madrid (UAM), Madrid, Spain  [2]Molecular Pathophysiology of Fibrosis, Physiological and Pathological Processes Program, Centro de Biología Molecular Severo Ochoa, Consejo Superior de Investigaciones Científicas (CSIC)/Universidad Autónoma de Madrid (UAM), Madrid, Spain [3]Microenvironment and Metastasis Group, Molecular Oncology Program, Spanish National Cancer Research Centre (CNIO), Madrid, Spain  [4]Vascular Biology and Therapeutics Program, Yale University School of Medicine, New Haven, CT, USA  [5]Integrative Cell Signalling and Neurobiology of Metabolism Program, Department of Comparative Medicine, Yale University School of Medicine, New Haven, CT, USA  [6]Instituto de Investigación Médica Mercedes y Martín Ferreyra (INIMEC)–Consejo Nacional de Investigaciones Científicas y Técnicas (CONICET)–Universidad Nacional de Córdoba (UNC), Córdoba, Argentina

Correspondence: fguix@cbm.csic.es; fguixrafols@gmail.com; cdotti@cbm.csic.es

 **Life Science Alliance**

(sEVs) and large EVs (Witwer & Théry, 2019). Exosomes, exomeres, and other vesicles with a size around 100 nm are considered sEVs. Microvesicles (200 nm–1 µm), apoptotic bodies (1–5 µm), and oncosomes (1–10 µm) are considered large EVs (Witwer & Théry, 2019). Exosomes are formed by invagination towards the lumen of the limiting membrane of multivesicular bodies (MVBs), a subtype of late-endosomal compartments appearing during the maturation of early endosomes (Raposo & Stoorvogel, 2013). During the invagination process, small vesicles with a size ranging from 50 to 150 nm, named intraluminal vesicles (ILVs), are generated, trapping cytosolic proteins and nucleic acids inside them. Lipids such as ceramide and cholesterol have been reported to play an important role in ILVs biogenesis (Simons & Raposo, 2009). MVBs can fuse with degradative compartments (autophagosomes or lysosomes), what leads to the degradation of ILVs, or alternatively, fuse with the plasma membrane and release ILVs to the extracellular space in form of exosomes. Previous work revealed that only MVBs with high cholesterol content fused to the plasma membrane and released exosomes (Möbius et al, 2003).

In light of the crucial role of this mechanism for the maintenance of cell function, we investigated the mechanisms of EV generation in aging neurons.

# Results

### Rodent neurons in culture as an experimental model for the study of the cell biology of degradation-secretion balance during aging

Studying cells in vitro is one of the experimental strategies used to generate knowledge on the fundamental biochemical and molecular processes underlying cell function, both in physiological and pathological conditions. With the similar objective of generating knowledge, many laboratories have used and use cells growing in culture to learn about the mechanisms that regulate death/survival equilibrium during aging (Cristofalo & Stanulis, 1978; Linskens et al, 1995; Pawelec et al, 2000; Acun et al, 2019; Hou et al, 2019). We and several other laboratories (Sodero et al, 2011; Guix et al, 2012; (Bigagli et al, 2016); Palomer et al, 2016; Ungureanu et al, 2016; Ishikawa & Ishikawa, 2020) use long-term primary cultures of rodent (mouse, rat) neurons as cellular model to learn about mechanisms of neuronal differentiation, survival, and plasticity in control and disease-like conditions. These studies are facilitated by the good understanding we have of the different phases that embryo-derived neurons (i.e., cortical, hippocampal) go through once seeded in culture. Thus, we know that differentiation and growth of axons and dendrites occurs during the first week, establishment of synaptic contacts and electric communication during the second week, wear and tear signs of stress develop during the third week, and signs of degeneration and death by apoptosis during the fourth week. Previous work from our laboratory demonstrated the gradual appearance of signs of aging/stress by the end of the third-week in culture and greater during the fourth week: that is, presence of lipofuscin granules, accumulation of reactive oxygen species, hyperphosphorylation of τ, increased stress/anti-stress activity (Akt, BCL2, pJNK, p53, and p21), reduced expression of synaptic plasticity genes, reduced synaptic activity,

impaired insulin signalling, altered plasma membrane composition, and reduced activity of the proteasome/autophagosome systems (Martin et al, 2008, 2011, 2014; Sodero et al, 2011; Trovò et al, 2013; Palomer et al, 2016; Benvegnù et al, 2017; Moreno-Blas et al, 2019; Martín-Segura et al, 2019b). Therefore, these cells carry out their entire life cycle in the course of 4 wk, with the fourth week showing, consistent with this lifespan, features of dysfunction typically found in vivo aging studies.

### Neurons aging in vitro show abnormal lysosomal function

Consistent with what occurs in various cell types during aging, both in vivo and in vitro (Cuervo & Dice, 2000; Terman & Brunk, 2004; Cuervo, 2008; Ishikawa & Ishikawa, 2020), here we show that also cortical neurons in vitro develop signs of lysosomal dysfunction in the last week of their 4-wk-long lifespan. Transmission electron microscopy (TEM) of neurons maintained in culture for 3 wk revealed larger and higher number of heterolysosomes than 1–2-wk-old neurons (Fig 1A–C), as well as higher number of the lysosome-related multilamellar bodies (MLBs) (Fig 1A and B), suggesting that aging in vitro has impaired the capacity of the degradative compartments (lysosomes and/or autophagosomes) to properly catalyse the turnover of aged or damaged membranes and aggregated proteins (see the accumulation of poly-ubiquitinated proteins after 3 wk in vitro (Fig S1A); Reviewed by De Araujo et al [2020]). Consistent with this possibility, analysis of LAMP2A as measurement of chaperone-mediated autophagy (Cuervo & Dice, 2000) significantly decreases with age in culture (Fig 1D), similarly to what occurs in vivo (Cuervo & Dice, 2000). On the other hand, the neurons in culture in which the effects of age are observed are alive, as reflected by the lack of increased lactate dehydrogenase release or protein loss (Fig S1B and C), indicating that also in the in vitro condition, mechanisms are activated that act as a buffer to reduce the damage that would be produced by the proteostasis defects.

### Rat cortical neurons produce more sEVs as they age in culture

In different cell types, a decrease in degradative capacity is compensated by an increase in the secretion of EVs, to eliminate part of the accumulated non-degraded material (Strauss et al, 2010; Miranda & Di Paolo, 2018; Papadopoulos et al, 2018; Gabandé-Rodríguez et al, 2019). To determine if this mechanism is activated in aging neurons, we quantified secreted EVs (sEVs) with Nanosight, a nanoparticle-tracking system device, from neurons kept in culture for different periods (schematic representation of EVs isolation procedure is shown in Fig S2). Nanosight quantification of the vesicles released in the culture medium (a typical size distribution profiles obtained with Nanosight and electron microscopy image of sEVs are shown in Fig 2A and B) revealed higher number of vesicles in the medium of neurons that had been kept in vitro for 3–4 wk compared with that of neurons maintained in culture for 2 wk (Fig 2C). Our cultures consist of always plating the same number of cells in serum-free medium, suggesting that the increase we see with age in vitro is genuine because of greater release by neurons and not to increased proliferation of non-neuronal cells with age. In further support of a neuron-aging

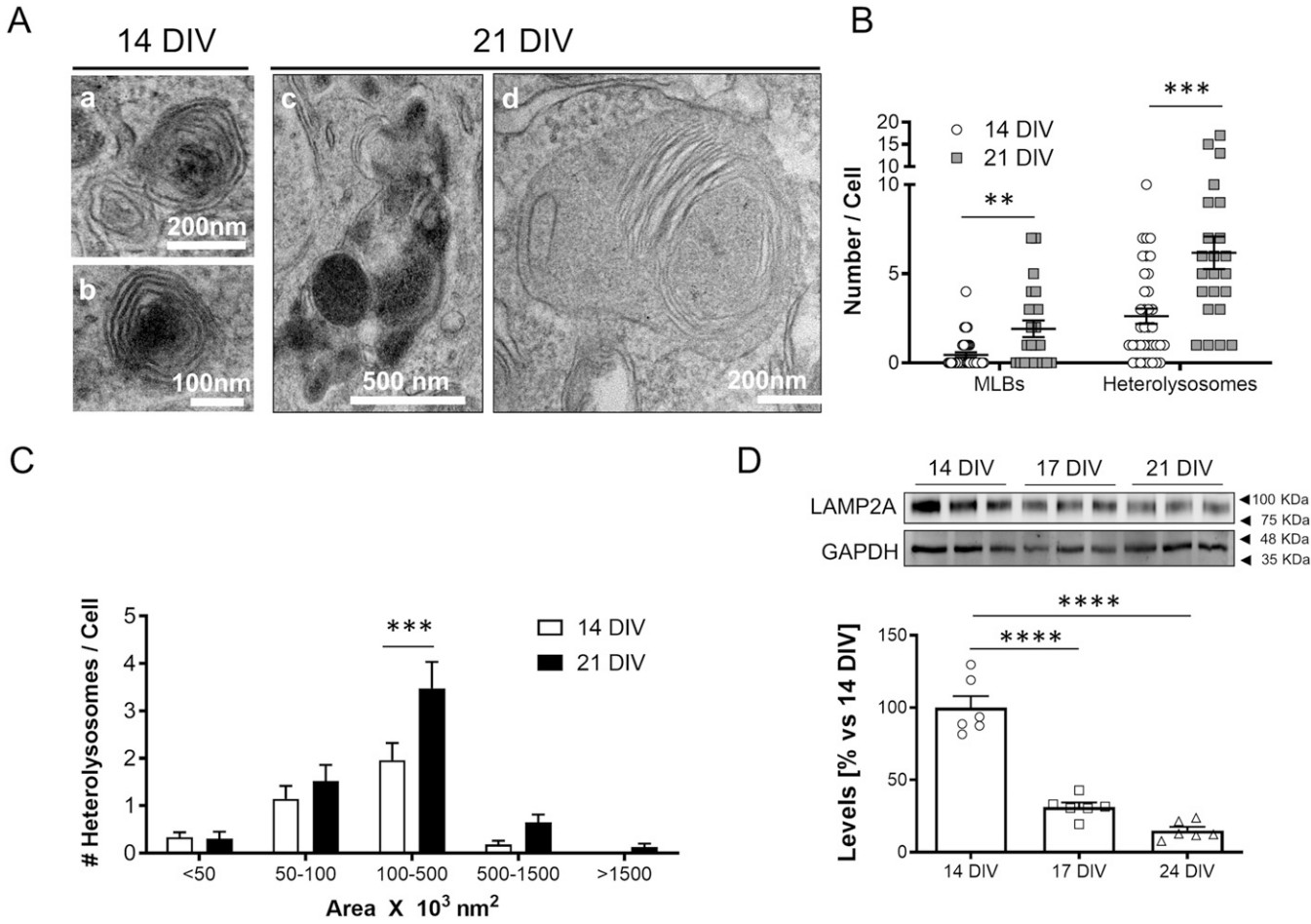

**Figure 1. Changes in degradative function with age in cultured neurons.**

**(A)** Representative transmission electron microscopy (TEM) images showing secondary lysosomal structures (heterolysosomes [Aa and Ac] and multilamellar bodies [Ad]) in 14 DIV (Aa, Ab) or 21 DIV (Ac, Ad) neurons. **(A, B)** Plot with the average number of multilamellar bodies (MLBs) and heterolysosomes per cell, calculated from TEM images such as the ones shown in panel (A) (n = 36 cells for 14 DIV and MLBs; n = 37 cells for 14 DIV and heterolysosomes; n = 23 cells for 21 DIV and both MLBs and heterolysosomes, N = 3 independent cultures). The graph shows the mean ± SEM. Statistical significance was analyzed by the two-tailed Mann–Whitney test (**$P < 0.01$, ***$P < 0.001$; $P$-value for MLBs: 0.0025; $P$-value for heterolysosomes: 0.0004). **(C)** Plot showing the distribution of heterolysosomes per cell grouped by ranges of areas (×$10^3$ nm²), calculated from TEM images such as the ones shown in panel a (n = 98 heterolysosomes for 14 DIV; n = 140 heterolysosomes for 21 DIV; N = 3 independent cultures). The graph shows the mean ± SEM. Statistical significance was analyzed by two-way ANOVA ($P$-value for interaction = 0.0289). Post hoc analysis was performed with Sidak's multiple comparisons test (***$P < 0.001$; $P$-value for 100–500 nm³ × $10^3$ nm²: 0.0002). **(D)** Western blot analysis of LAMP2A as measurement of chaperone-mediated autophagy with age in culture. Note that LAMP2A protein levels decrease with age in vitro, from 14, 17 and 21 DIV. An anti-GAPDH antibody was used to normalize the LAMP2A protein levels. A graph with the LAMP2A levels relative to 14 DIV neuronal culture is shown under the Western blot panels (n = 6 independent cultures). The graph shows the mean ± SEM. Statistical significance was analyzed by one-way ANOVA ($P < 0.0001$). Post hoc analysis was performed with Dunnet's multiple comparisons test (****$P < 0.0001$ for all comparisons).

associated process, the production of sEVs increases daily as neurons age in culture (Fig 2D). The analysis by Nanosight also revealed a slight decrease in the size of sEVs with neuronal aging in vitro (mean ± 95% Confidence Interval): 144.9 ± 6 nm for 12–14 DIV; 135.4 ± 6.35 nm for 15–21 DIV; 126.8 ± 11.65 nm for 22–29 DIV.

Because of the fact that Nanosight performs poorly at detecting vesicles under 50 nm in size, we quantified smaller vesicles (<50 nm) released to the medium by TEM. A higher number of vesicles under 50 nm in size were also found in the media of 3-wk-old in vitro neurons in comparison to 2-wk-old cultures (Figs 2E and S3A–D). The vast majority of vesicles fall in the range of 50–150 nm in size, the typical size of exosomes (mean percentage of vesicles ± 95% confidence

interval: 65.59% ± 4.09% for 12–14 DIV; 71.8% ± 4.55% for 15–21 DIV; 77.69% ± 5.03% for 22–29 DIV. Biochemical analysis of the sEVs by Western blot confirmed the presence of the exosomal markers HSP90 and CD81, as well as τ protein for sEVs isolated from the medium of 3-wk-old neuronal cultures (Figs 2F and S4A–D), indicating that although the increased secretion of vesicles can help to detoxify the old neurons, these same vesicles can be harmful to neighbouring cells (see the Discussion section).

Because exosomes are released to the extracellular environment after fusion of late endosomes/multivesicular bodies (MVBs) with the plasma membrane, we reckoned that the release of more exosomes by older neurons would be mirrored by more numerous or larger multivesicular bodies (MVBs) with more intralumenal

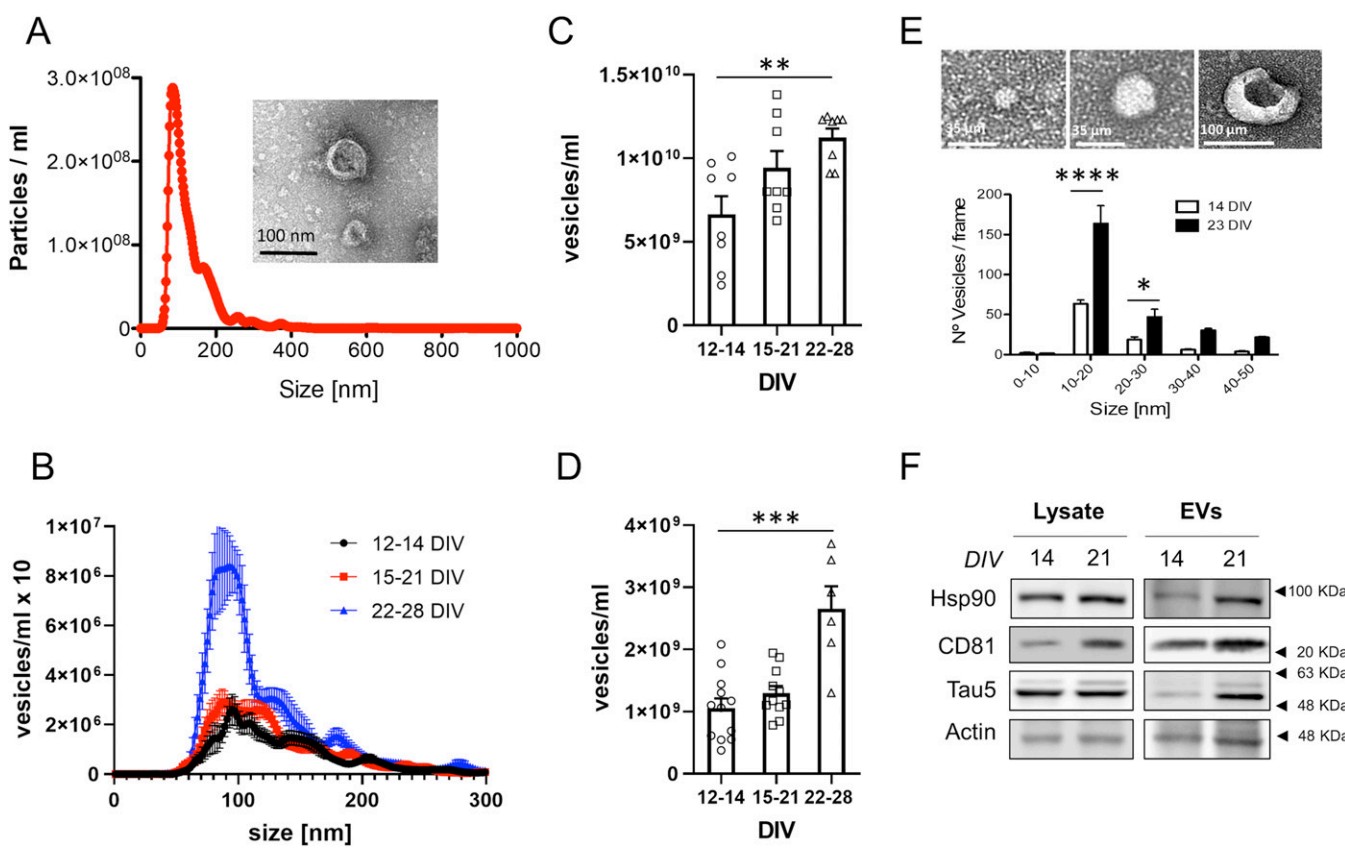

**Figure 2. Characterization of extracellular vesicles secreted by young and old neuronal cultures.**
**(A)** Representative plot showing the size distribution of the vesicles present in the medium of a rat neuronal culture and analyzed by Nanosight (determined in the supernatant obtained after centrifugation at 10,000$g$). The profile shows that most of the vesicles have a size between 50–200 nm. The insert shows a representative transmission electron microscopy (TEM) image of vesicles obtained from the medium of rat neuronal cultures by centrifugation at 100,000$g$. **(B)** The plot compares the size distribution of vesicles present in the medium of 12–14, 15–21 or 22–28 DIV neuronal cultures, determined by Nanosight (n = 11 for 12–14 DIV; n = 11 for 15–21 DIV; n = 6 for 22–28 DIV). **(B, C)** The plot compares the concentration of vesicles present in the accumulated medium of 12–14, 15–21 or 22–28 DIV neuronal cultures, quantified from the area under the curve of size distribution curves obtained by Nanosight, such as the one shown in panel (B) (n = 8 per group). The graph shows the mean ± SEM. Statistical significance was analyzed by Kruskal–Wallis test (*P*-value: 0.0164). Post hoc analysis was performed with Dunn's multiple comparisons test (**P* < 0.05; *P*-value for comparison of 12–14 DIV and 22–28 DIV: 0.0083). **(D)** The same analysis as in C but for small extracellular vesicles secreted during a 24-h period. (n = 12 for 12–14 DIV; n = 11 for 15–21 DIV; n = 6 for 22–28 DIV). The graph shows the mean ± SEM. Statistical significance was analyzed by one-way ANOVA (*P* < 0.0001). Post hoc analysis was performed with Dunnet's multiple comparisons test (****P* < 0.0001; *P*-value for comparison of 12–14 DIV and 22–28 DIV <0.0001). **(E)** TEM images showing three examples of vesicles observed in the medium of neuronal cultures (analyzed in the pellet obtained by centrifugating the medium pooled from three independent cultures at 100,000$g$). Below, the plot compares the number of vesicles by size groups (0–50 nm), between 14 DIV and 23 DIV. The quantification was obtained from the analysis of TEM images such as the ones shown above. The graph shows the mean number of vesicles ± SEM. Statistical significance was analyzed by two-way ANOVA (*P*-value for interaction <0.0001). Post hoc analysis was performed with Sidak's multiple comparisons test (**P* < 0.05; ****P* < 0.0001; *P*-value for comparison of 10–20 nm between 12–14 DIV and 22–28 DIV: *P* < 0.0001; *P*-value for comparison of 20–30 nm between 12–14 DIV and 22–28 DIV: 0.0375). **(F)** Western blot characterization of total lysates and small extracellular vesicles isolated from the medium of 14 and 21 DIV neuronal cultures by ultracentrifugation at 100,000$g$. The blots were analyzed with antibodies against exosomal markers (HSP90 and CD81), a total tau antibody (Tau5) and an antibody against actin as a loading control.

vesicles (ILVs). Quantitative electron microscopy analysis confirmed this prediction (Fig 3A–D).

### Accumulation of cholesterol in endocytic organelles of cortical neurons aging in culture is due to NPC1 down-regulation

The degradation defects, lysosome enlargement (Fig 1B and C), and increased exosome production (Fig 2) observed in aging cortical neurons in culture reproduce some of the features found in cells of patients, and animal models, with Niemann Pick type C (NPC) disease. In this condition, the proteostasis phenotype (i.e., autophagic-lysosomal dysfunction) is due to a loss-of-function

mutation in the cholesterol-transporter NPC1 protein (in a minority of cases the mutation is in the NPC2 protein) resulting in the abnormal accumulation of cholesterol and GM2 in late endocytic/MVBs and lysosomes (Zervas et al, 2001). Hence, we tested the possibility that the proteostasis changes in the aging neurons in culture are the consequence of a similar cholesterol accumulation defect. Cholesterol levels and distribution were measured in neurons of different age in vitro using BODIPY-labelled cholesterol as a cholesterol sensor. Figs 4A and S5 show that while BODIPY-cholesterol is diffusely distributed in the cytoplasm of 2-wk-old neurons, it becomes more abundant (see quantification in Fig 4B) and concentrated in structures positive

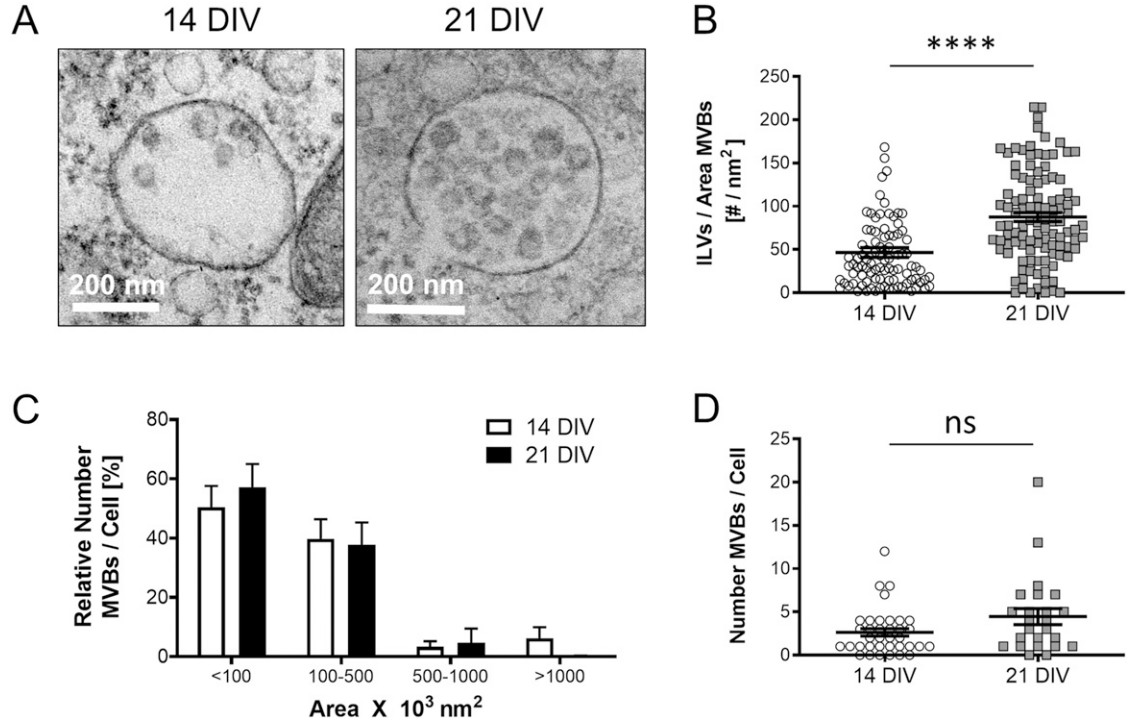

**Figure 3. Analysis of multivesicular bodies (MVBs) in cortical neurons during aging in vitro.**
**(A)** Representative transmission electron microscopy (TEM) images of 14 DIV and 21 DIV cortical neurons showing an increased number of ILVs in the lumen of MVBs in 21 DIV neuronal cultures when compared with 14 DIV. **(A, B)** The plot compares the number of ILVs normalized by the area (nm$^2$) of the MVBs, quantified from TEM images of 14 DIV and 21 DIV neuronal cultures, such as the ones shown in panel (A) (n = 94 MVBs for 14 DIV; n = 107 MVBs for 21 DIV; N = 3 independent neuronal cultures). The graph shows the mean ± SEM. Statistical significance was analyzed by two-tailed Mann–Whitney test (****<0.0001). **(A, C)** The plot compares the size distribution of MVBs between 14 DIV and 21 DIV neuronal cultures, calculated from TEM images such as the ones shown in panel (A) (n = 31 cells and 101 MVBs for 14 DIV; n = 21 cells and 110 MVBs for 21 DIV; N = 3 independent neuronal cultures). Relative (%) MVBs numbers per each size group and cell are calculated by comparing to the total number of MVBs present in the cell. The graph shows the mean ± SEM. Statistical significance was analyzed by two-way ANOVA (P-value for interaction: 0.7081, ns: non-significant). **(D)** The plot compares the total number of MVBs per cell between 14 DIV and 21 DIV neuronal cultures. **(A)** The quantification was done on TEM images such as the ones shown in panel (A) (n = 36 cells for 14 DIV; n = 23 cells for 21 DIV; N = 3 independent cultures). The graph shows the mean ± SEM. Statistical significance was analyzed by two-tailed Mann–Whitney Test (ns: non-significant; P-value: 0.1078, ns).

for markers of early endosomes (EEA1) in neurons kept in culture for more than 3 wk. This is even more evident with the marker of late endosomes/lysosomes LIMP-1 (Figs 4A and S6 and quantified in Fig 4C).

To explore whether the accumulation of cholesterol in endocytic structures in older neurons in culture is due to an age-associated reduction of NPC1, we next measured the levels of this protein by Western blot. Fig 4D shows that NPC1 becomes significantly down-regulated with time in vitro, starting in the third week. In addition, the levels of another cholesterol transport protein, ABCA1, are also reduced in 3–4-wk-old neuronal cultures compared to 2-wk-old cultures (Fig 4E).

To directly assess if age-associated NPC1 down-regulation in the cultured neurons plays a role in the increased MVB/sEVs phenotype, we carried out two types of experiments: inactivation of NPC1 function and reduced its expression.

For the inactivation, we treated neurons with the drug U18666A, which specifically binds and inactivates the sterol-sensing domain of this protein (Lu et al, 2015) and is the pharmacological tool of choice to mimic NPC1 loss of function in Niemann Pick C disease

studies (Lange et al, 2000; Lu et al, 2015; Vivas et al, 2019). Consistent with a direct cause–effect relationship, 2-wk-old neurons in culture incubated with U18666A showed cholesterol accumulation in endocytic structures in contrast with untreated cells (Fig S7A). Moreover, the treated neurons also exhibited MVBs with more ILVs in their lumen (Fig 5A and B) and an increased release of EVs (Figs 5C and D and S7B and C).

Decreasing the levels of the NPC1 protein through transduction of 2-wk-old neurons with a lentivirus expressing an shRNA against the mRNA of NPC1 also led to the increase in the number of ILVs (Fig 5E–G). In further support of a cause–effect association, over-expressing NPC1 in old neurons led to a reduction in the number of ILVs in the lumen of MVBs (Fig 6A–C) without any other noticeable alteration in other organelles (e.g., mitochondria). Although this methodology did not allow us to determine the effect on exosome secretion (only a proportion of the cultured cells over-express NPC1) the observations that acute reduction of NPC1 phenocopies chronic, age-associated, NPC1 reduction (compare data in Figs 2 and 3 with 5), and that over-expression rescues the endosomal phenotype (Fig 6A–C) support the contention that the

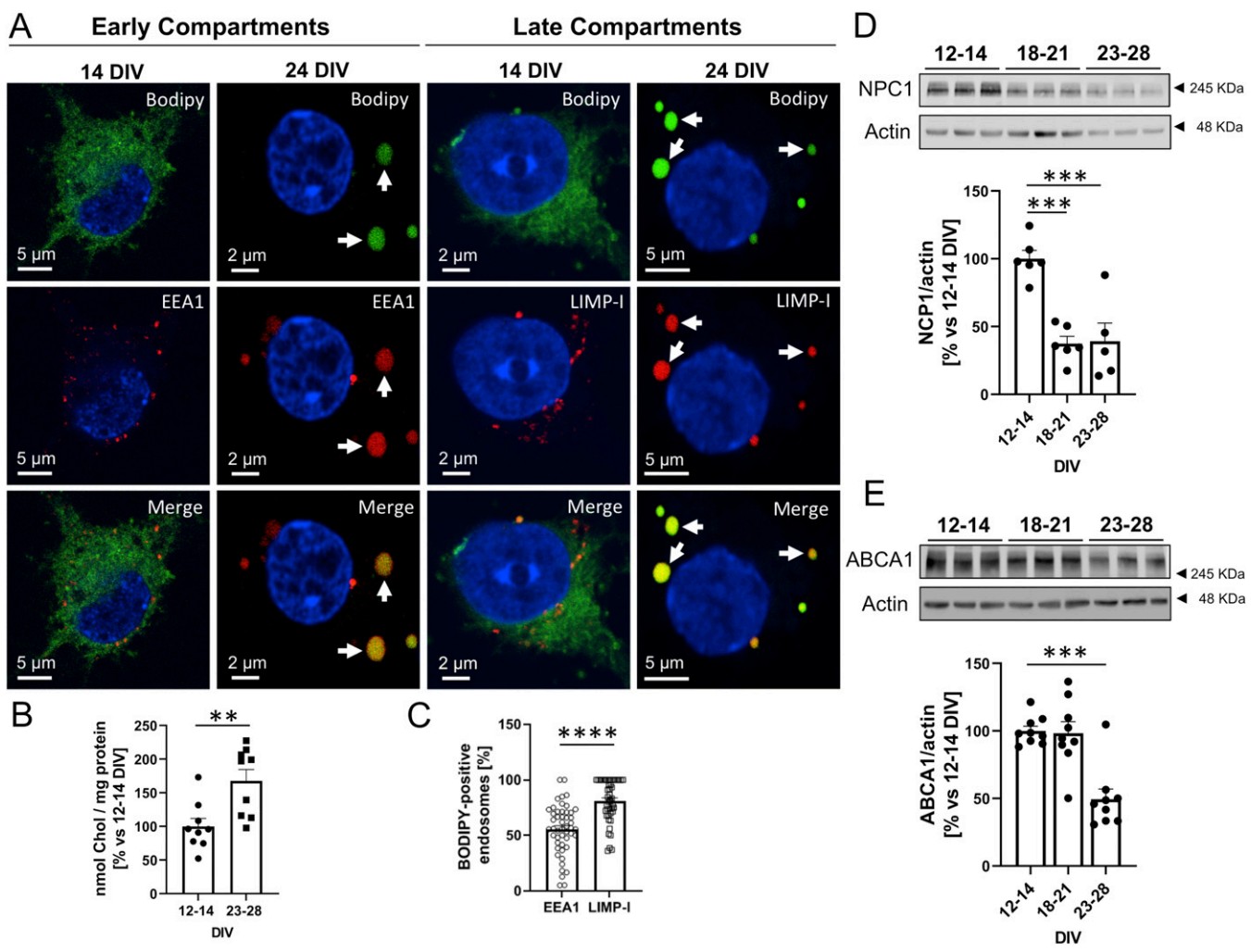

**Figure 4. BODIPY-cholesterol accumulates in endosomal compartments in rat primary cortical neurons aged in vitro.**
**(A)** Representative confocal microscopy images of 14 and 24 DIV rat primary cortical neurons treated with 1 μM BODIPY-cholesterol (green) for 24 h and stained with antibodies against an early endosomal marker (EEA1, red), which stains endosomal compartments where internalized material ends up first, or a late endosomal/lysosomal marker (LIMP-I, red), which stains a mature state of endosomal compartments. 24 DIV neurons accumulate BODIPY-cholesterol in early and late-compartments (white arrows). **(B)** The plot compares the relative amount (% versus 12–14 DIV) of BODIPY-cholesterol taken by 12–14 DIV and 23–28 DIV primary cortical neurons in a 24 h period, calculated by the intracellular fluorescent signal (Ext 495 nm/Em 507) of BODIPY-cholesterol (nmol of cholesterol where from a BODIPY-cholesterol standard curve) and normalized by the total protein in the sample (% versus 12–14 DIV; n = 9 independent cultures). The graph shows the mean ± SEM. Statistical significance was analyzed by two-tailed unpaired $t$ test (**$P < 0.01$; $P$-value: 0.0046). **(A, C)** Plot showing the percentage of the total EEA1-positive (n = 1,536) or LIMP-I positive (n = 1,269) compartments that accumulate BODIPY-cholesterol, calculated from confocal images as the ones shown in panel (A) (N = 3 independent experiments). The graph shows the mean ± SEM. Statistical significance was analyzed by two-tailed Mann–Whiitney Test (****$P < 0.0001$). **(D)** Western blot analysis of total lysates of 12–14 DIV, 18–21 DIV, and 23–28 DIV rat cortical neurons shows a pronounced decrease in NPC1 protein levels in 18–21 DIV neurons that is maintained in 23–28 DIV neurons. An antibody against beta-actin was used to normalize the protein levels. Below, a plot with the quantification of the bands of Western blot experiments as the one shown. Protein levels are relative to 12–14 DIV. The graph shows the mean ± SEM. Statistical significance was analyzed by one-way ANOVA ($P$-value: 0.0001). Comparison of the age-groups 18–21 DIV and 23–28 DIV to 12–14 DIV was performed by Dunnet's multiple comparisons tests (n = 6 for 12–14 DIV and 18–21 DIV; n = 5 for 23–28 DIV, ***$P < 0.001$; $P$-value for 18–21 DIV to 12–14 DIV comparison: 0.0002; $P$-value for 23–28 DIV to 12–14 DIV comparison: 0.0004). **(E)** Western blot analysis of lysates of neuronal cultures at different aging points. The blot was tested with an antibody against the cholesterol transporter ABCA1 and actin. Below, a plot compares the relative ABCA1 protein levels between 12–14 DIV, 18–21 DIV and 23–28 DIV neuronal cultures. The bands corresponding to ABCA1 were quantified from blots such as the one shown above. Statistical significance was analyzed by Kruskal–Wallis Test ($P$-value 0.0022). Comparison of the age-groups 18–21 DIV and 23–28 DIV to 12–14 DIV was performed by Dunn's Test (n = 9 independent lysates per group; **$P < 0.01$; $P$-value for 23–28 DIV to 12–14 DIV comparison: 0.0036).

down-regulation of NPC1 with aging is involved in the neurons' EVs increased release.

### NPC1 down-regulation in cortical neurons aging in culture is initiated by an Akt/mTORC1-degradation mechanism

In cancer cell lines activation of the protein kinase B (Akt)/mammalian target of rapamycin (mTOR) signalling pathway causes

proteasomal-mediated degradation of NPC1 (Du et al, 2015). Thus, we tested to which extent NPC1 down-regulation in our cultured neurons could be due to age-associated Akt/mTOR activation. Western blot analysis of extracts of neurons cultured for different lengths of time revealed that Akt activity slowly increases around the end of the second week, and that this increase is paralleled both by the increase in the levels of the phosphorylated form of the mTOR substrate S6 Kinase (S6K) and the decrease in the levels of

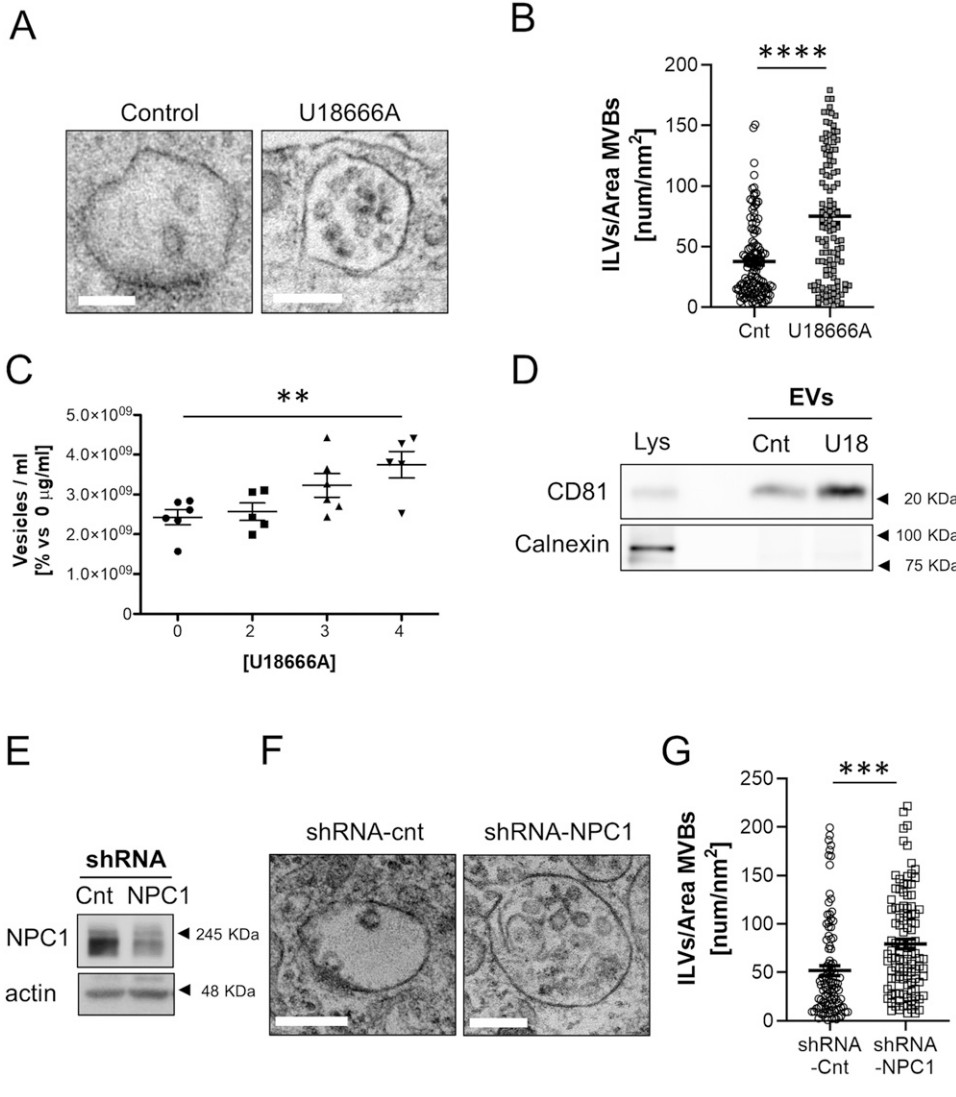

**Figure 5. NPC1 inhibition in 14 DIV cortical neurons recapitulates the alterations observed in neurons aged in vitro.**
**(A)** Representative transmission electron microscopy (TEM) images of multivesicular bodies (MVBs) from untreated (control) or 4 µg/µl U18666A-treated 14 DIV cortical neurons. Scale bars = 200 nm. **(A, B)** Plot showing the number of ILVs normalized by the area (nm²) of the MVB that contains them, between untreated (control) or 4 µg/µl U18666A (U18)-treated 14 DIV neuronal cultures, quantified from TEM images such as the ones shown in panel (A) (n = 121 MVBs; N = 2 experiments). The graph shows the mean ± SEM. Statistical significance was analyzed by two-tailed Mann–Whitney Test (****$P$ < 0.0001). **(C)** The graph shows the quantification of extracellular vesicles by Nanosight from the medium of untreated 14 DIV neurons (control) or treated with increasing concentrations of U18666A. The graph shows single experiments, the mean ± SEM. Statistical significance was analyzed by one-way ANOVA ($P$-value: 0.009). Comparison of 4 µg/µl U18666A (U18)-treated neurons to control group was performed by Dunnet's multiple comparisons test (**$P$ < 0.001; n = 6 for control and 3 µg/µl; n = 5 for 2 and 4 µg/µl). **(D)** Western blot analysis of a total lysate of 14 DIV neuronal culture and total lysates of small extracellular vesicles isolated from the medium of 14 DIV untreated (Cnt) or U18-treated neurons. The blot was tested with an antibody against CD81, a marker for exosomes, and calnexin (an ER protein used as a negative marker for exosomes). **(E)** Western blot experiment of total lysates of 14 DIV neuronal cultures transduced with a lentivirus expressing a scrambled shRNA as a control (Cnt) or an shRNA against NPC1. Blots were probed with an antibody against NPC1 or actin as a loading control. **(F)** Representative TEM images of 14 DIV neuronal cultures transduced with a scrambled shRNA (shRNA-cnt) or an shRNA against NPC1 (shRNA-NPC1). **(G)** The plot compares the average number of ILVs normalized by the area (nm²) of the MVB containing them, between shRNA-cnt and shRNA-NPC1 transduced neurons. **(F)** Data were obtained by quantification of TEM images as the ones shown in panel (F) (n = 100 MVBs for shRNA-cnt; n = 108 for shRNA-NPC1; N = 2 independent cultures). The graph shows the mean ± SEM. Statistical significance was analyzed by two-tailed Mann–Whitney test (****$P$ < 0.0001).

NPC1 (Fig 7A–C). To test if there is a cause–effect relationship between the two types of biochemical changes, we increased Akt activity, either by adding Insulin Growth Factor 1 or the pan-Akt activator SC79. Either of these two conditions led to a significant increase in Akt phosphorylation and the significant reduction in NPC1, both in the neuron-like N2A cells and in primary cortical neurons (Fig 7D and E). In further agreement, inhibiting Akt activity led to a marked increase in NPC1 in N2A and primary neurons (Fig 7F and G). To explore whether Akt-mediated reduced NPC1 in old neurons could be due to increased mTOR activity, the levels of NPC1 were analyzed in cells incubated with various concentrations of the mTORC1 inhibitor Rapamycin. In support of a mTORC1 effect, rapamycin treatment increased NPC1 levels (Fig 7H) and decreased secretion of sEVs (Fig 7I). Interestingly, blocking NPC1 in young neurons decreases mTOR activity (Fig S8A and B), what may work as a negative loop to regulate the mTOR-induced degradation of NPC1.

## NPC1 down-regulation with age is maintained by an miR-33 inhibition mechanism

Akt levels begin to decline after the third week, but NPC1 levels remain low, suggesting that an extra mechanism of NPC1 inhibition is activated with aging.

One mechanism involved in the control of cholesterol homeostasis implies the action of the microRNA 33 (miR-33) in the introns of the sterol regulatory element-binding protein (SREBP) genes *Srebf*-1 and *Srebf*-2 (Marquart et al, 2010) which code for the cholesterol synthesis and transport regulatory transcription factors SREBP-1 and SREBP-2 (Bommer & MacDougald, 2011). Under conditions of low cholesterol or low NPC1 protein levels, the NPC1-facilitated cholesterol translocation from lysosomes to ER (Höglinger et al, 2019) is decreased and SREBP-2 is activated, what enhances the transcription of its target genes. Because the *Srebf2* gene has a sterol regulatory element sequence on the promoter (Miserez et al, 1997),

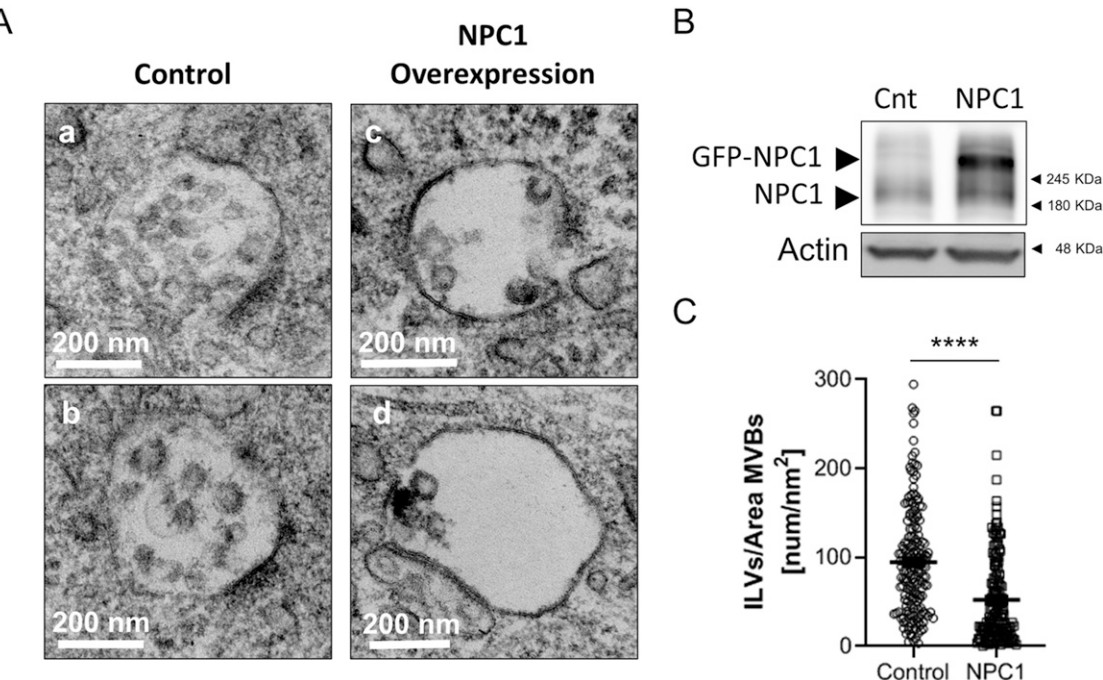

**Figure 6.   NPC1 over-expression reduces ILVs in aged neurons.**
**(A, C, D)** Representative transmission electron microscopy images of multivesicular bodies (MVBs) from non-infected 21 DIV neurons (control; subpanels [a and c]) or infected (NPC overexpression, subpanels [c and d]) with a Sindbis virus expressing GFP-tagged NPC1 (GFP-NPC1). **(B)** Western blot analysis of total lysates of 21 DIV control neurons (cnt) or overexpressing GFP-NPC1 (NPC1). Blots were tested with an antibody against NPC1 and Actin. **(C)** The plot compares the number of ILVs normalized by the area of the MVBs that contained them (nm$^2$), between control and GFP-NPC1-overexpressing (NPC1) neurons. **(A)** Quantifications were performed on transmission electron microscopy images such as the ones shown in panel (A) (n = 181 MVBs for control; n = 161 MVBs for NPC1; N = 3 independent cultures). The graph shows the mean ± SEM. Statistical significance was analyzed by two-tailed Mann–Whitney test (****$P$ < 0.0001).

SREBP-2 activation enhances also its own transcription. miR-33a is co-transcribed with *Srebf-2* and inhibits the transcription of the ATP binding cassette (ABC) cholesterol transporters ABCA1 and ABCG1, as well as the NPC1 thus reducing cholesterol efflux (Rayner et al, 2010). Thus, we wondered whether the levels of miR-33 mRNA are increased in 3-wk-old neuronal cultures compared with 2-wk-old-cultures, and if this increase could be responsible for maintaining low levels of NPC1. qPCR analysis revealed that miR-33 mRNA levels are significantly increased in 3-wk-old neuronal cultures compared with 2-wk-old cultures (Fig 8A). In support of a direct causal link with NPC1 down-regulation, over-expression of an miR-33 mimics plasmid caused the down-regulation of NPC1 and ABCA1 (Fig 8B–D). Finally, to demonstrate the importance of this microRNA in the regulation of neuronal levels of NPC1, we analyzed the levels of NPC1 in brain samples from miRNA33 KO mice. In the miR33 KO condition, NPC1 and ABCA1 protein levels are increased, although the increase in ABCA1 does not reach statistical significance because of one of the mice of the WT group (Figs 8E and F and S9A and B).

## Discussion

In this work, we show that age-associated down-regulation of the cholesterol transport protein NPC1 may operate as double-faced, gain-loss, mechanism. On the one hand, there is the negative/loss

effect. Different studies have demonstrated that the loss of function of NPC1, either by mutations or by reduction in its levels, leads to the accumulation of cholesterol in organelles of the endo-lysosomal pathway resulting in functional alterations, although the pathogenic agent in NPC1 seems to be the accumulation of the ganglioside GM2 (Zervas et al, 2001). The same studies have shown that the accumulation of cholesterol in these organelles interferes with their hydrolytic capacity (Liao et al, 2007) (Reviewed by Futerman and van Meer [2004]) responsible in the end for the development of the neurological and psychiatric alterations typical of this disease (Imrie et al, 2015). However, the cultured neurons live for more than a week after the observed accumulation of cholesterol, suggesting that the negative consequences of NPC1 down-regulation are, in the cultured neurons, compensated. In this sense, cholesterol accumulation in MVBs appears to act as a compensatory event because it is required for the formation of membrane tethers and ILVs (future exosome) formation (Boura et al, 2012; Eden et al, 2016; Höglinger et al, 2019). As NPC1 levels become reduced and cholesterol accumulates in endosomes of early and, mainly, late endocytic organelles (Fig 3), more membrane tethers and vesicles form, remaining retained in the lumen of MVBs, in the form of ILVs (Fig 2). In fact, cholesterol is highly enriched in ILVs harbouring 85% of all the cholesterol found in MVBs (Möbius et al, 2003), and endosomal accumulation of cholesterol in oligodendrocytes due to NPC1 blockage has been shown to boost exosome production (Strauss et al, 2010). Consistent with this view, we showed that increasing NPC1 in old neurons led to MVB with less ILVs,

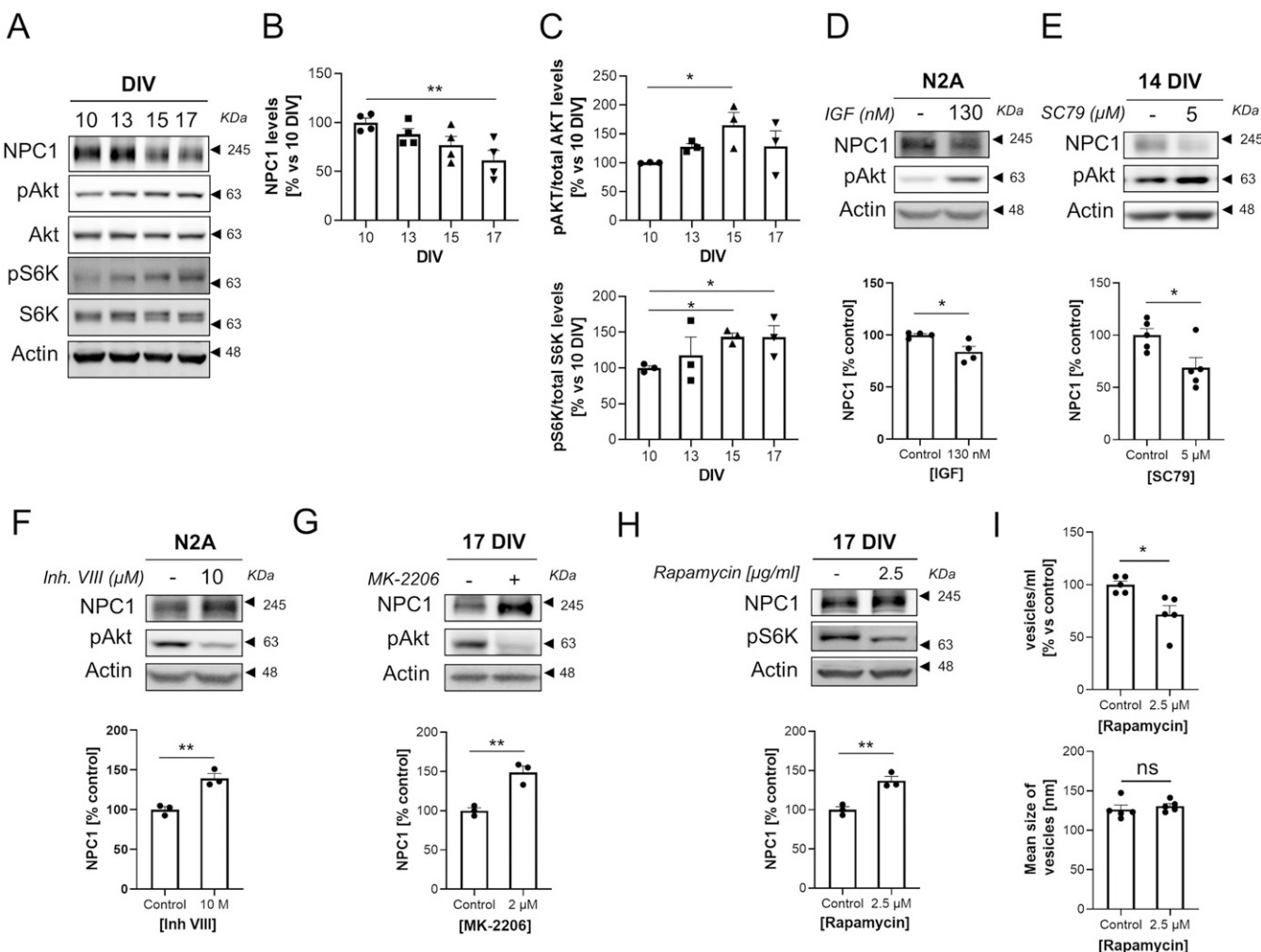

**Figure 7. Akt activation during neuronal aging in vitro triggers NPC1 degradation.**
**(A)** Western blot analysis of total lysates of neuronal cultures at different aging points (10, 13, 15 and 17 DIV). The blot was tested with antibodies against NPC1, Akt phosphorylated at serine 473 (pAKT), total Akt, the mTOR substrate S6 Kinase (S6K) phosphorylated at threonine 389 (pS6K), total S6K, and actin. **(A, B)** The plot compares the levels of NPC1 normalized to actin, during neuronal aging in vitro. The protein levels were quantified from the bands of Western blot experiments as the ones shown in panel (A) (n = 4 independent cultures). The graph shows the mean ± SEM. The means for 10, 13, 15 and 17 DIV were significantly different when compared by ANOVA (*P*-value: 0.0219). Dunnet's multiple comparisons tests were used for the post hoc analysis of the data (**P* < 0.01; *P*-value for the 10 DIV to 17 DIV comparison: 0.0049). **(C)** Alterations of the AKT-mTORC1 pathway during neuronal aging in vitro. **(A)** The plots show changes in the levels of pAKT (normalized to total AKT; upper plot) and pS6K (normalized to total S6K; lower plot) at the same aging points of neuronal cultures as in panel (A). **(A)** The protein levels were quantified from the bands of Western blot experiments as the ones shown in panel (A) (n = 3 independent cultures). The graph shows the mean ± SEM. The means for 10, 13, and 15 DIV (for pAKT, upper blot) and the means for 10, 15 and 17 DIV (for pS6K, lower blot) were significantly different when compared by ANOVA (*P*-value for pAKT: 0.0308; *P*-value for pS6K: 0.0320). Dunnet's multiple comparisons tests was used for the post hoc analysis of data (**P* < 0.05; *P*-value for 15 DIV to 10 DIV comparison in pAKT: 0.0199; *P*-value for 15 DIV to 10 DIV comparison in pS6K: 0.0362; *P*-value for 17 DIV to 10 DIV comparison in pS6K: 0.0370). **(D)** Western blot analysis of the total lysates of N2A cells untreated or treated with 130 nM insulin growth factor for 24 h. The blot was tested with antibodies against NPC1, pAKT (ser473) and actin. Below, a plot comparing the NPC1 levels between untreated and Insulin Growth Factor-treated N2A cells. Quantifications were carried out on Western blot experiments as the one shown above (n = 4 independent cultures). The graph shows the mean ± SEM. Statistical significance was analyzed by two-tailed unpaired *t* test (**P* < 0.05; *P*-value: 0.0231). **(E)** Western blot analysis of total lysates of 14 DIV neuronal cultures treated with DMSO (control) or 5 μM of the pan-Akt activator SC79 for 24 h. The blot was tested with antibodies against NPC1, pAKT (ser473), and actin. Below, a plot comparing the NPC1 levels between DMSO-treated and SC79-treated neurons. Quantifications were carried out on Western blot experiments as the one shown above (n = 5 independent cultures). The graph shows the mean ± SEM. Statistical significance was analyzed by two-tailed unpaired *t* test (**P* < 0.05; *P*-value: 0.027). **(C, F)** N2A cells treated with DMSO (control) or Akt inhibitor VIII for 48 h were analyzed as in panel (C) (n = 3 independent cultures). The graph shows the mean ± SEM. Statistical significance was analyzed by two-tailed unpaired *t* test (**P* < 0.01; *P*-value: 0.0049). **(F, G)** Lysates of 17 DIV cortical neurons treated with DMSO (control) or the selective Akt inhibitor MK-2206 (2 μM) for 48 h were analyzed by Western blot with the same antibodies as in panel (F). Below, a plot comparing the NPC1 levels between DMSO-treated and MK-2206-treated neurons (n = 3 independent cultures). Statistical significance was analyzed by two-tailed unpaired *t* test (**P* < 0.01; *P*-value: 0.0052). **(H)** Lysates from 17 DIV cortical neurons treated with DMSO (control) or the mTOR inhibitor rapamycin (2.5 μM) were analyzed by Western blot. The blot was tested with antibodies against NPC1, pS6K (thr389), and actin. Below, the plot compares the NPC1 levels between DMSO-treated and 2.5 μM rapamycin-treated neurons. Quantification of the bands were carried out on blots such as the ones shown above (n = 3 independent cultures). The graph shows the mean ± SEM. Statistical significance was analyzed by two-tailed unpaired *t* test (**P* < 0.01; *P*-value: 0.0053). **(I)** Comparison of the concentration (upper plot) and size (lower plot) of vesicles in the medium of 48 h DMSO (Cnt)- or 2.5 μl/ml rapamycin-treated 18 DIV neuronal cultures determined by Nanosight (n = 5 independent cultures). The graph shows the mean ± SEM. Statistical significance was analyzed by two-tailed unpaired *t* test (**P* < 0.05, ns: non-significant; *P*-value for concentration: 0.0134; *P*-value for size: 0.5072).

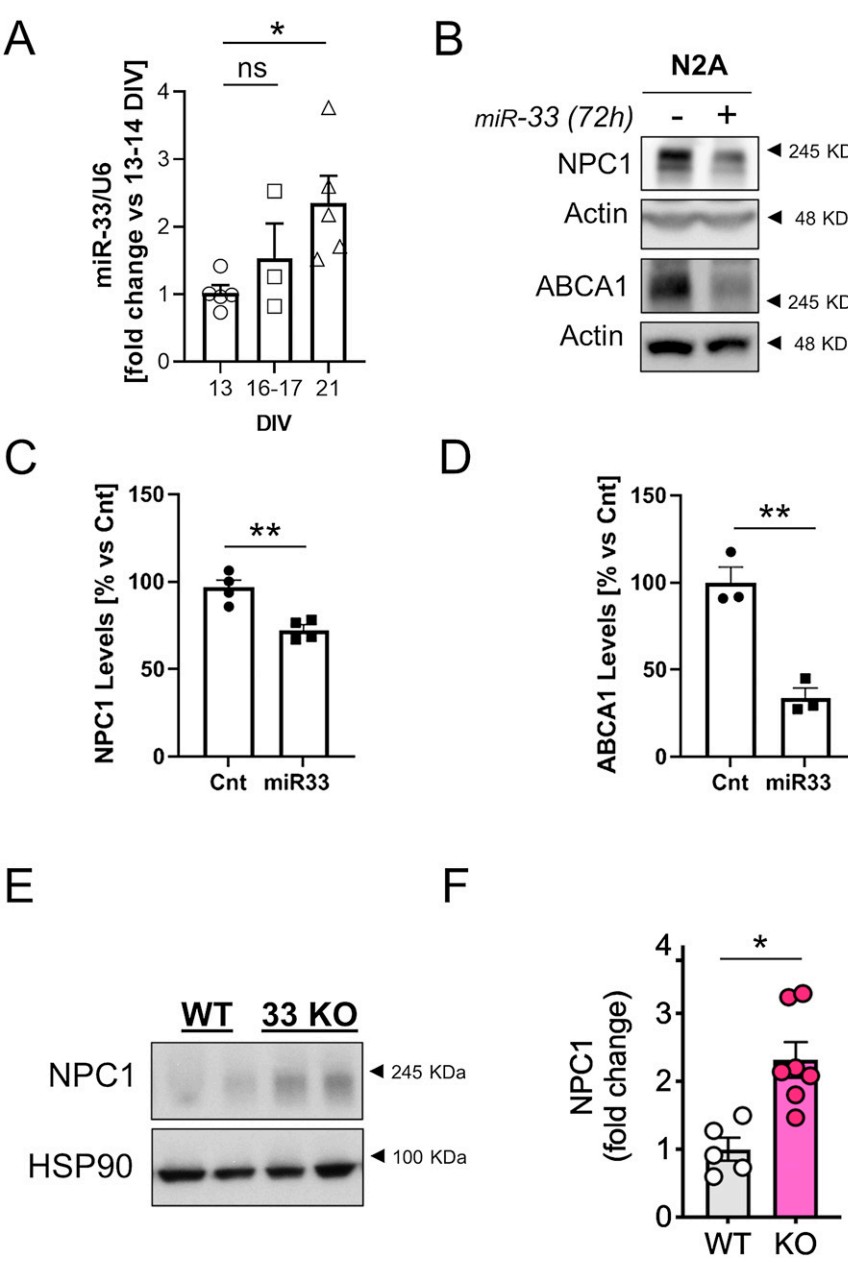

**Figure 8. miR33 is up-regulated in aged neurons in vitro.**
**(A)** The plot compares the miR33 levels (normalized by the housekeeping RNA U6) between 13 DIV, 16–17 DIV and 21 DIV neuronal cultures (fold change versus DIV). The microRNA levels were determined by RT–PCR (n = 5 independent culture). The graph shows the mean ± SEM. Statistical significance was analyzed by one-way ANOVA (*P*-value: 0.0410). Dunnet's multiple comparisons test was used to compare the 21 DIV and the 16–17 DIV to the 13 DIV group (ns: non-significant; *P < 0.05; *P*-value for the 21 DIV to 13 DIV comparison: 0.0258). **(B)** Western blot analysis of lysates of N2A cells transfected with 40 nM of a non-targeting control mimic or 40 nM miR-33 mimic (miRNA-33). The blots were analyzed with antibodies against NPC1, ABCA1 and actin. **(C)** Plot comparing the relative protein levels of NPC1 between N2A cells transfected with 40 nM of a non-targeting control mimic (n = 4) or 40 nM miR33 mimic (miR-33; n = 4). **(B)** Blots were quantified from Western blot experiments as the one shown in panel (B). The graph shows the mean ± SEM and individual data points. Statistical significance was analyzed by two-tailed unpaired *t* test (**P < 0.05). **(C, D)** The plot compares the ABCA1 protein levels in the same experimental conditions as in panel (C). The graph shows the mean ± SEM and individual data points (n = 3). Statistical significance was analyzed by two-tailed unpaired *t* test (**P < 0.05). **(E)** Western blot analysis of brain lysates from 2 mo old WT or miR33 KO mice. Blots were tested with antibodies against NPC1 and HSP90. **(F)** Plots comparing the relative protein levels for NPC1 in the brain of WT (n = 5) and miR33 KO (n = 7) mice. **(C)** Protein levels were quantified from Western blot experiments as the one shown in panel (C). The graph shows the mean ± SEM and individual data points. Statistical significance was analyzed by two-tailed unpaired *t* test (*P < 0.05).

whereas inhibiting NPC1 in young neurons resulted in MVBs filled with ILVs. Therefore, the reduction in NPC1 that can lead to impaired degradative function because of excess cholesterol is also responsible for generating more ILVs/exosomes, therefore a potential protective mechanism. The protective role of increasing exosome secretion has been evidenced on numerous occasions, mainly in pathological situations including neurodegeneration (Rajendran et al, 2006; Perez-Gonzalez et al, 2012; Deng et al, 2017; Patel et al, 2017; Guix et al, 2018; Fussi et al, 2018; Miranda et al, 2018; Ferreira et al, 2019; Burillo et al, 2021). Supporting this role in our culture system, we showed that exosomes secreted by the older neurons in culture are enriched in the microtubule associated protein τ (Fig S4), therefore reducing the risk of formation of stable intracellular oligomers and

tangles inside the old neurons that should seriously disrupt neuronal function (Shafiei et al, 2017). On the other hand, these same exosomes may exert toxic effects on neighbouring cells or even at a distance. In keeping, exosomes have been proposed as one of the mechanisms by which toxic proteins are spread throughout the brain in neurodegenerative conditions (Kalani et al, 2014; Perez et al, 2019). In fact, it is well known that τ proteins present in exosomes can be internalized by other neurons (Wang et al, 2017), leading to their aggregation and alteration of host cell function (Asai et al, 2015; Polanco et al, 2016). Thus, as with the NPC1 depletion mechanism leading to "damage-and-healing" events, the exosome response bears also double-edge effects, beneficial for the cells that release them but potentially deleterious for those that receive their cargo. Interestingly

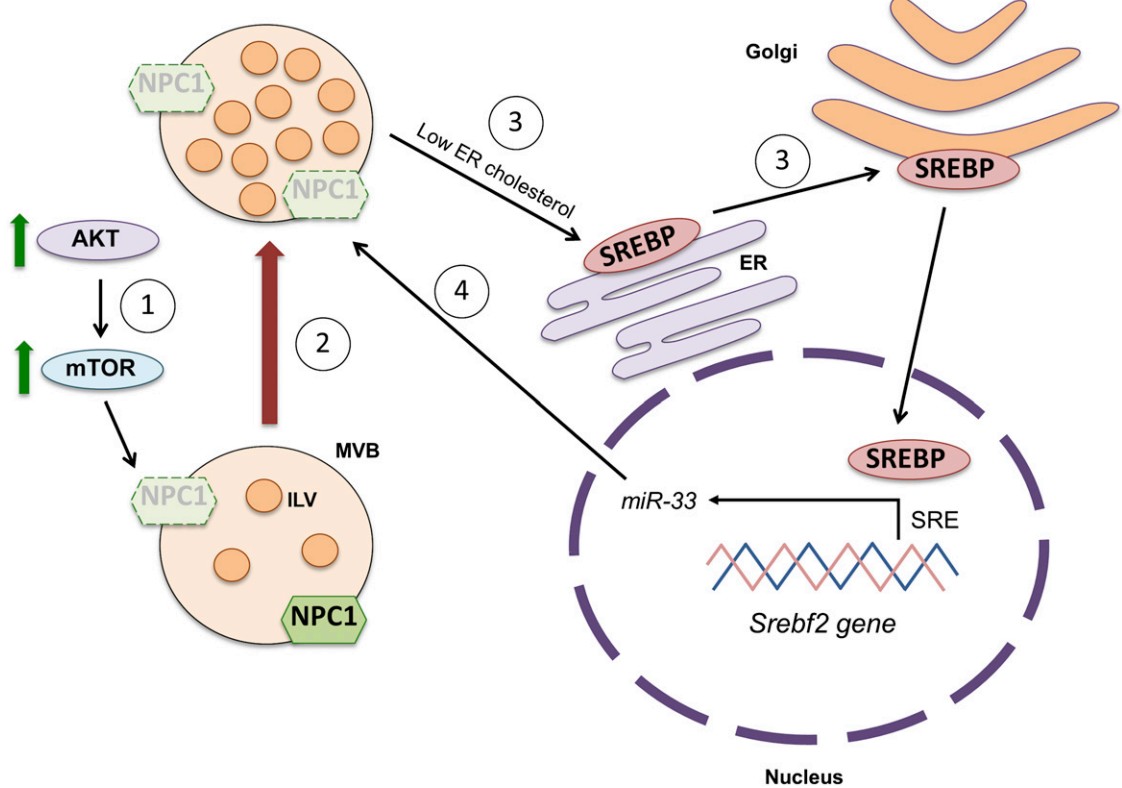

**Figure 9. Scheme showing the main alterations found in aging neurons in vitro.**
(1) The activation of the AKT-mTOR pathway during neuronal aging triggers the degradation of the intracellular cholesterol transporter Niemann Pick disease, type C protein (NPC1), what induces the accumulation of cholesterol in endosomal compartments. (2) This in turn induces multivesicular bodies (MVBs) to generate a higher number of intraluminal vesicles (ILVs, called exosomes when they are secreted) in their lumen. (3) Decreased levels of NPC1 provoke a low endosome-to-ER cholesterol transport, what triggers the activation of the Sterol regulatory element-binding proteins through the Golgi and its translocation to the nucleus. There, sterol regulatory element-binding protein activates the transcription of the microRNA miR-33 located in the intronic region of the Srebf2 gene, after binding its sterol response element (SRE). (4) miR-33 blocks the synthesis of NPC1 by targeting its mRNA, what aggravates the accumulation of endosomal cholesterol and the formation of ILVs.

enough, τ-positive neurofibrillary tangles have been detected in the brain of Niemann Pick patients with mutations in the *npc1* gene (Love et al, 1995; Vanier & Suzuki, 1998).

Another question that our work helps to answer has to do with the mechanism that leads to the decline in NPC1 with age. Our data are consistent with the possibility that the decrease in NPC1 in the old neurons can be due to an increase in degradation, to a decrease in synthesis, or to both. Degradation of NPC1 occurs in the lysosomes and in the proteasome, the latter mediated by mTORC1 (Du et al, 2015; Schultz et al, 2018). Previous work from our laboratory revealed that old neurons in culture present high mTORC1 activity due, in part, to high PI3K/Akt activity (Martín-Segura et al, 2019a). In this work we show that either Akt or mTORC1 activation are sufficient to reduce NPC1 levels, supporting the possibility that these pathways may be responsible, at least in part, for the reduction in NPC1 levels in old neurons (Fig 9). Although the accumulation of cholesterol in the endo-lysosomal system could also activate mTORC1 activity (Castellano et al, 2017), this mechanism seems not to contribute to maintain NPC1 at low levels after the third week in culture, when Akt levels start to decrease since inhibition of NPC1 in young neurons decreases mTOR activity in our cultures.

But in addition to the increased activity of the Akt and mTORC1 pathways in the brain during physiological aging (Martín-Segura et al, 2019a), the increased expression of the microRNA miRNA33 can contribute to maintaining low levels of NPC1. As explained in the previous section, the accumulation of cholesterol in the endo-lysosomal pathway would result in reduced levels of cholesterol in the ER, cleavage and activation of SREBPs, transcription of the *Srebf2* gene and the co-expression of microRNA 33 (miR-33) and repression of NPC1 synthesis. The fact that we observed age associated decrease in NPC1 in the cultured neurons and that the expression of a miR-33 mimicking construct reduces NPC1 strongly supports a functional link between the age-associated up-regulation of this microRNA and NPC1 down-regulation (Fig 9). Naturally, the data in KO mice reinforce the importance of the regulation of this miRNA in the control of expression of NPC1. Hence, we propose that cholesterol accumulation with age may elicit concerted or independent mechanisms (Akt, mTORC1, and miR-33) promoting NPC1 down-regulation in an effort to preserve neurons from the unwanted effects of dysfunctional proteostasis.

From a physiological perspective, the negative consequences on the old neurons' accumulation of cholesterol in organelles of the endocytic pathway as a consequence of the decrease in NPC1 are

counterbalanced by the function of the sequestered cholesterol in the formation of ILVs (Fig 9) and their release in the form of EVs. Thus, although a greater release of exosomes can alleviate neurons with proteostasis problems (i.e., removing undegraded proteins and membranes) the cells that receive this type of cargo may themselves be harmed, contributing to the irreversibility of the aging process. Proposed strategies of increasing removal of intracellular waste as a therapeutic avenue in old age should be cautiously considered.

# Materials and Methods

## Isolation of extracellular vesicles

The 24-h conditioned medium (25 ml of MEM medium with N2 supplement) collected from one 150-mm culture dish per condition was subsequently centrifuged at 200$g$ and 2,000$g$ for 10 min each to remove dead cells and debris. After transferring the supernatant to a fresh tube (Cat. no. 344058; Beckman), 1× protease inhibitors were added (cOmpleteTM; Sigma-Aldrich) and the final volume was made up to 38 ml with filtered PBS/1× protease inhibitors. Next, the supernatant was centrifuged at 10,000$g$ and 4°C for 30 min in a TST28.38 rotor (Kontron) to remove microvesicles. The supernatant was transferred to a fresh tube (Cat. no.344058; Beckman) and centrifuged at 100,000$g$ and 4°C for 2 h in a TST28.38 rotor (Kontron). The pellet was washed with 0.5 ml filtered PBS/1× protease inhibitors and centrifuged at 100,000$g$ and 4°C for 1 h in a TLA100.1 rotor (Beckman). Finally, the pellet was resuspended in filtered PBS 1× for EM analysis, or RIPA lysis buffer (see formulation in the Western blot analysis section) with 1× protease and phosphatase inhibitors for Western blot analysis.

## Rat primary cortical cultures

Cortical neurons obtained from rat embryos present numerous morphological and functional characteristics of cortical neurons in situ, both morphological (pyramidal neurons) and in the type of synaptic contacts they make (excitatory) (Dichter, 1978). Primary cultures of cortical neurons were prepared from embryonic day 18 (E18) Wistar rats as described in the study by Kaech and Banker (2006). Cortex was dissected and placed into ice-cold Hank's solution (Hanks Buffer Salt Solution: Ca2+ and Mg2+ free; Thermo Fisher Scientific) supplemented with 7 mM Hepes and 0.45% glucose. The tissue was then treated with 0.005% trypsin (trypsin 0.05% EDTA; Invitrogen; Life Technologies Co.) and DNase (72 mg/ml; Sigma-Aldrich) incubated at 37°C for 16 min. Cortex was washed three times with Hank's solution. Cells were dissociated in 5 ml of plating medium (Minimum Essential Medium supplemented with 10% horse serum and 20% glucose) and cells were counted in a Neubauer chamber. Cells were plated into dishes pre-coated with 0.1 mg/ml poly-D-lysine (Sigma-Aldrich), 4.8 × 10$^6$ cells in a 150 mm dish, and 300,000 cells/well in a six multi-well plate and afterwards, they were placed into a humidified incubator containing 95% air and 5% CO2. The plating medium was replaced with equilibrated neurobasal media supplemented with B27 and GlutaMAX (Gibco; Life Technologies Co.) after 4 h.

**Information on the antibodies used in this study.**

| Target | Company | Catalogue number | Host | Dilution |
| --- | --- | --- | --- | --- |
| ABCA1 (H1J) | Novus Biologicals | NB100-2068 | Rabbit | 1:2,000 |
| Akt | Cell Signaling Technology | 9272S | Rabbit | 1:1,000 |
| Calnexin | Abcam | Ab22595 | Rabbit | 1 μg/ml |
| CD81 | Santa Cruz Biotechnology | sc-166029 | Mouse | 1:200 |
| EEA1 | BD Bioscience | 610457 | Mouse | 1:200 |
| GAPDH | Abcam | Ab8245 | Mouse | 1:5,000 |
| HSP90 | Sigma-Aldrich | 05–594 | mouse | 2 μg/ml |
| Lamp2a | Invitrogen | 51–2,200 | Rabbit | 1:500 |
| LIMP-I | Given by Prof, Ignacio Sandoval | | Mouse | 1:300 |
| NPC1 | Novus Biologicals | NB400-148 | Rabbit | 1:2,500 |
| p(Thr389)-S6K | Cell Signaling Technology | 9234S | Rabbit | 1:1,000 |
| p-Akt (S473) | Cell Signaling Technology | 4060S | Rabbit | 1:1,000 |
| Akt | Cell Signaling Technologies | 9272S | Rabbit | 1:1,000 |
| S6K | Cell Signaling Technology | 2708S | Rabbit | 1:1,000 |
| Secondary goat anti-mouse | DAKO | P0447 | Goat | 1:2,500 |
| Secondary goat anti-rabbit | DAKO | P0448 | Goat | 1:2,500 |
| Tau5 | Invitrogen | AHB0042 | mouse | 1:500 |
| Tubulin | Sigma-Aldrich | T5168 | Mouse | 1:4,000 |
| β-Actin | Abcam | Ab8227 | Rabbit | 1:5,000 |

On DIV 8, the culture medium was progressively replaced (1/4 of the media each day for 3 d, the fourth day being totally replaced) with MEM media supplemented with N2 (Gibco; Life Technologies Co.) and without GlutaMAX. Cortical neurons were kept in culture the days corresponding to each condition.

## Colocalization of BODIPY-cholesterol accumulation with endosomal markers

14 DIV or 24 DIV, rat primary cortical neurons were seeded on six-well plates (300,000 neurons per well) containing coverslips pre-coated with poly-D-lysine (0.5 mg/ml poly-D-lysine in borate buffer; Sigma-Aldrich). 24 h before fixation, 1 μM BODIPY-cholesterol (TopFluor Cholesterol [23-dipyrrometheneboron difluoride-24-norcholesterol], Cat. no. 810255; Avanti Lipids) was added to the medium. Cells were fixed in 4% PFA diluted in PBS for 10 min at room temperature and at dark. Then, PFA was replaced by PBS 1× and samples were kept at 4°C until immunofluorescence analysis. The day of the experiment, cells were permeabilized for 5 min with 0.1% Triton X-100 in PBS 1×, then blocked for 15 min with 3% BSA in PBS 1×. Next, coverslips were incubated with the primary antibodies diluted in PBS 1× for 2 h: a mouse monoclonal anti-EEA1 antibody (1:200; BD Bioscience, Cat. no.610457) or a mouse monoclonal anti-Limp I antibody, given by Prof. Ignacio Sandoval (1:300). Next, coverslips were washed three times with PBS 1×, and incubated for 1 h with an Alexa 555–conjugated anti-mouse antibody (Thermo Fisher Scientific). After a 5 min DAPI staining (1 μg/ml in PBS 1×; Merk), and three washes with PBS 1×, coverslips were finally fixed using Mowiol gel mounting agent (DABCO).

The preparations were analyzed with an LSM 710 confocal microscope system (Zeiss). Zeiss imaging software was used to analyze the confocal images.

## Cholesterol accumulation assay

1 μM BODIPY-cholesterol (TopFluor Cholesterol 23-dipyrromethene-boron difluoride-24-norcholesterol, #810255; Avanti Polar lipids, inc) was added to rat primary cortical neurons seeded on six-wells plates (300,000 neurons per well). 24 h later, cells were washed three times with PBS 1× and scraped in 200 μl/well of PBS 1×. Each sample was transferred to a well of a 96-wells plate and fluorescence was measured in a spectrophotometer (PerkinElmer) (Excitation 488 nm/Emission 510). The concentration of the BODIPY-cholesterol present in cells was calculated by a standard curve consisting of BODIPY-cholesterol diluted in PBS 1× at the following known concentrations (nM): 2,500, 1,250, 625, 312.5, 156.25, 78.125, 39.063, 19.532, 9.766, 4.883, 2.442, and 0 as a blank. The total BODIPY-cholesterol amount was calculated taken into account the volume and normalized by the total protein present in the sample, which was determined by a BCA assay (Pierce BCA Protein Assay kit; Thermo Fisher Scientific; see the Determination of protein concentration section in Materials and Methods).

## N2A cell culture

Murine neuroblastoma cell line (N2A) cells were grown in DMEM supplemented with 10% FBS and 100 IU/ml penicillin, and 100 mg/ml

streptomycin (complete DMEM). Cells were incubated at 37°C, humidity conditions, and 5% $CO_2$.

For the test to probe NPC1 and ABCA1 to be a target of miR-33, N2A cells were transfected with 40 nM miR-33 mimic using RNAiMAX (Thermo Fisher Scientific) for 8 h. Experimental control samples were treated with an equal concentration of a nontargeting control mimic sequence. 72 h after transfection, cells were lysed in RIPA buffer with complete protease inhibitors for Western blot analysis.

## Cell treatments

N2A or rat primary cortical neurons were plated on six multi-well plates (pre-coated with 0.1 mg/ml poly-D-lysine (Sigma-Aldrich) in the case of neurons) at a density of 300,000 cells/well. The following compounds were added to cell media of primary cultures or N2A cells, for the duration and concentration indicated in each experiment: U18666A NPC1 inhibitor (#U3633; Sigma-Aldrich); BODIPY-cholesterol (TopFluor Cholesterol 23-dipyrrometheneboron difluoride-24-norcholesterol, #810255; Avanti Polar lipids, inc), human peptide purified Insulin Growth Factor 1 (#78022; Stem Cell Technologies); Akt inhibitor VIII (#124018; Calbiochem); SC79 Akt activator (#SML0749; Sigma-Aldrich); and MK-2206 (Cat# 11593; Cayman); Rapamycin (#R8781; Sigma-Aldrich).

## Western blot analysis

Cells or EVs were lysed in RIPA buffer (20 mM Tris–HCl, pH 7.5, 150 mM NaCl, 1 mM EDTA, 1 mM EGTA, 1% NP–40, 1% sodium deoxycholate, and 0.1% SDS) with phosphatase inhibitors (Sigma-Aldrich) and protease inhibitors (cOmpleteTM; Sigma-Aldrich). Proteins were prepared in Laemmli buffer (Tris–HCl, 25 mM, pH 6.8, SDS 1%, glycerol 3.5%, 2-mercaptoethanol 0.4%, and bromophenol blue 0.04%) and separated by electrophoresis in polyacrylamide gels in the presence of SDS at constant voltage. Subsequently, they were transferred onto nitrocellulose membranes and after blocking with blocking solution (1% BSA in 0.1% Tween-20 in TBS [T-TBS]), membranes were incubated with the corresponding primary antibody (Table 1 below) diluted in blocking buffer overnight at 4°C. After washing the membranes with T-TBS, they were incubated with the relevant secondary antibodies coupled to horseradish peroxidase and diluted 1/2,500 for 1 h at RT. The proteins recognized by the antibodies were detected with luminol (Pierce ECL Western Blotting Substrate; Thermo Fisher Scientific), and chemiluminescence was measured using a charge coupled device camera (Amersham Imager 680). The bands corresponding to the proteins of interest were densitometrated by the FIJI digital image processing software and were normalized with respect to the values obtained for the charge control protein actin.

## Determination of protein concentration

The concentration of proteins present in the homogenates was determined by means of the BCA assay (Pierce BCA Protein Assay kit; Thermo Fisher Scientific), following the indications of the commercial kit.

## Nanoparticle tracking analysis

1 ml of the conditioned medium (MEM supplemented with N2 supplement) of rat primary cortical neurons seeded in a six multi-well plate

(300,000 cells/well; 1.5 ml per well), was subsequently centrifuged at 200$g$ and 2,000$g$ for 10 min each to remove dead cells and debris. The supernatant was transferred to a fresh Eppendorf tube and centrifuged at 10,000$g$ and 4°C for 30 min. The supernatant was diluted 1:10 in filtered PBS 1× to obtain a concentration in the range of 1–10 × 10$^8$ vesicles/ml, and the concentration of vesicles present in the sample was analyzed by Nanoparticle Tracking Analysis by using a NanoSigh NS500 instrument (Malvern Instruments). The instrument was equipped with a 488-nm laser, a high-sensitivity complementary metal-oxide-semiconductor camera, and a syringe pump. Both the concentration of the vesicles present in the media of cortical neurons at different DIV, as well as the mean size, were analyzed using the nta2.3 software (Malvern Instruments) after filming three 60-s videos.

### TEM

Cell cultures were processed by conventional inclusion in epoxy resin (TAAB 812). First, the media is discarded and the cells are fixed with 4% paraformaldehyde + 2% glutaraldehyde in phosphate buffer 0.1 M, pH 7.4, for 2 h at RT. Next, cells are washed three times with phosphate buffer 0.1 M, pH 7.4, for 10 min. Then, the samples are subjected to post-fixation treatment. First, cells are treated with osmium tetroxide 1% in bidistilled water plus 1% of phosphate ferricyanide for 1 h at 4°C, followed by three washes in bidistilled water, 5 min each. Then, tannic acid 0.15% in phosphate buffer was added for 1 min at RT, followed by three washes, the first in the same buffer and the second and third in bidistilled water, 5 min at RT each one. Finally, treatment with uranyl acetate 2% in water for 1 h and in darkness, followed by three washes in oxygenated water. Afterwards, the samples were dehydrated with consecutive baths in ethanol at different concentrations, 50, 75, 90, 95, and 3 × 100%, 5–10 min at 4°C each time. Then, inclusion was performed by successive treatments with the epoxy resin EPON and ethanol in different proportions. First, EPON: ethanol 1:2, then 1:1, and finally 1:2, for 60 min at RT each time. A final treatment with EPON100% was left overnight at 4°C. Finally, the minimum quantity of complete EPON was added and left to polymerize for 48 h at 60°C. Next, cuts of 70 nm approximately were obtained in the microtome with a diamond knife, and sections were placed in Cu/Pd hexagonal gratings. Sections were stained with uranyl acetate 2% in water for 7 min and lead citrate for 2 min at RT. The microscope used was Jeol Jem-1010 (Jeol), and pictures were taken with the camera 4K × 4K F416 from TVIPS and the TEM images were analysed.

### RNA isolation

Total RNA extraction was performed using QIAzol Lysis Reagent following the miRNeasy Mini Kit (Cat. no.217004; QIAGEN) protocol.

### Quantification of microRNAs

Retrotranscription (RT) was performed as specified in miRCURY LNA RT Kit (Cat. no.339340; EXIQON) protocol from 10 ng of initial RNA. Quantitative RT–PCR (qRT-PCR) was carried out with the miRCURY LNA SYBR Green PCR Kit (Cat. no.339345; EXIQON), using miRNA LNA PCR primers for miR-33 (Ref. YP00205690; EXIQON) and miRNA LNA PCR primers for the housekeeping U6 (PER-YCP0050862; EXIQON) for miRNA expression analysis. qRT-PCR was performed in a 96-well Bio-Rad CFX96 RT–PCR System with a C1000 Thermal Cycler (Bio-Rad). A Ct value was obtained from each amplification curve using Bio-Rad CFX Manager Software.

All qRT-PCR reactions were performed in duplicate. Relative miRNA expression was determined using the 2−ΔΔCt method. Raw data from real time gene expression from technical replicates were averaged, expression of the experimental targets was normalized to the expression of the housekeeping U6, and fold change above 13 DIV neurons was then calculated.

### Generation of lentiviral vectors

All the necessary procedures to produce lentiviruses were carried out in a culture room with a P2 biosecurity level. To generate second-generation lentiviral particles, HEK 293T cells were transfected at 80% confluence with a mix of the following plasmids prepared in OptiMEM (Thermo Fisher Scientific) and containing the transfection agent polyethyleneimine (PEI; Sigma-Aldrich) in a PEI: DNA 1: 1 ratio: a packaging plasmid (pCMV delta R 8.2; Addgene), a shell plasmid VSV-G (pMD2.G; Addgene) coding for vesicular stomatitis virus glycoproteins, and the following plasmids of interest: a silencing plasmid expressing an shRNA against NPC1 (shRNA-NPC1; TL501501; OriGene) or a scrambled shRNA (shRNA-Cnt; TR30021; OriGene). After 12 h, the medium was replaced by DMEM with 2% FBS and the cells were maintained at 37°C for an additional 48 h. After this time, the medium was collected and centrifuged at 900$g$ for 15 min at 4°C to remove cell debris and filtered with a 0.22 nm pore filter (Millipore). Subsequently, the supernatant was ultracentrifuged at 60,000$g$ for 2 h at 4°C. The pellet corresponding to the lentivirus was resuspended in sterile PBS and frozen in single-use aliquots that were maintained at −80°C.

To estimate the concentration of lentiviruses expressed in Transducing Units /ml, HEK 293T cells were infected with increasing concentrations of virus and the percentage of infected cells (positive for GFP) after 48 h was obtained. Neuronal cultures were infected at six DIV with a MOI of 5. To do this, on the day of infection, the amount of lentivirus needed was resuspended in 200 $\mu$l of conditioned medium and added to the culture of neurons dropwise. The next day, 1/4 of the media was replaced with MEM media supplemented with N2 (Gibco; Life Technologies Co.) and without GlutaMAX, and each day for 3 d, the fourth day being totally replaced.

### EGFP-NPC1 overexpression

The plasmid containing EGFP-tagged NPC1 was purchased from Addgene (#53521). Then, the EGFP-NPC1 insert was re-cloned into the pSinRep5 vector for Sindbis virus production and infection of 19 DIV rat primary cortical neurons. Neurons were lysated at 21 DIV in RIPA buffer with phosphatase inhibitors (Sigma-Aldrich) and proteases inhibitors (cOmpleteTM; Sigma-Aldrich) and analyzed by Western blot (see the Western blot analysis section of Materials and Methods).

### MiR-33 knockout mice

Generation of miR-33−/− mice was accomplished with the assistance of Cyagen Biosciences Inc. CRISPR/Cas9-mediated excision of miR-33 was accomplished using targeted guide sequences toward intron 16 of the Srebp-2 gene. Establishment of the miR-33bKI mouse model was accomplished in collaboration

with ingenious Targeting Laboratory. To generate these mice, the entire mouse Srebp-1 gene was removed and replaced with the human sequence, including the region encoding miR-33b. The success of both of these approaches has been verified by southern blotting and confirmed by PCR-based genotyping using specific primers.

### Statistics and reproducibility

No statistical methods were used to calculate sample sizes, but our sample sizes are similar to those reported in previous publications (Song et al, 2016; Miranda & Di Paolo, 2018; Joshi et al, 2020). Although experimental conditions were not blinded, data analysis was performed blind whenever possible. The number of independent experimental repeats was indicated in figure legends. Data are shown as mean ± SEM and individual data points are presented except for Figs 2E and 3C, because of visualization clarity. For Fig 4C, data are presented with a box-and-whiskers plot showing the minimum and the maximum values. Data were analyzed using the GraphPad Prism 6 or Excel 2011 software. When data were not normally distributed, Mann–Whitney test (for comparison of two groups) or Kruskal–Wallis test (for comparison of three or more groups) were used to analyze the data. Dunn's multiple comparisons test was used for the post hoc analysis. For data normally distributed, two-tailed unpaired $t$ test (for comparison of two groups) or one-way/two-way ANOVA (for comparison of three or more groups) were used for the analysis. Dunnet's multiple comparisons test or Šidák's multiple comparisons tests were used for the post hoc analysis of data tested with one-way ANOVA or two-way ANOVA, respectively. $P$-values <0.05 are considered to be statistically significant. Whenever possible, the exact $P$-values are provided in figure legends.

### Ethics statement

Wistar rats were housed at the Centro de Biología Molecular "Severo Ochoa" (CSIC). All experiments were performed in accordance with European Union guidelines (2010/63/UE) regarding the use of laboratory animals.

For the miR-33 KO mice, animal studies were approved by the Institutional Animal Care and Use Committee of Yale University School of Medicine.

## Data Availability

All data that support the findings of this study are available from the corresponding authors upon request. This study includes no data deposited in external repositories.

## Supplementary Information

## Acknowledgements

We thank the Electron Microscopy Service of the Centro de Biología Molecular Severo Ochoa (http://www.cbm.uam.es/joomla-rl/index.php/es/investigacion/servicios-cientificos/microscopia-electronica) for staining and processing of samples, with a special acknowledgement for Milagros Guerra Rodríguez. We thank the laboratory of María Dolores Ledesma and Daniel N Mitroi for giving us the Sindbis virus for the EGFP-NPC1 overexpression experiments. This work was partially supported by the SAF2016-76722, PID2019-104389RB-I00 (AEI/FEDER, UE), EU JPND "EpiAD," and NextGeneration EU-CSIC funds (NeuroAging) to CG Dotti; a Ministerio de Economía y Competitividad (MINECO) PID2019-104233RB-100 and the European Regional Development Fund, Instituto de Salud Carlos III REDinREN RD16/0009/0016 and Comunidad de Madrid "NOVELREN" B2017/BMD-3751 to S Lamas. FX Guix was partially supported by a Marie Skłodowska-Curie Actions-Individual Fellowship (T2DM and AD, EU 708152).

### Author Contributions

FX Guix: conceptualization, data curation, formal analysis, funding acquisition, investigation, prepared the manuscript, and writing—review and editing.
AM Capitán: formal analysis, investigation, and run Western blot experiments and analyzed TEM images.
Á Casadomé-Perales: formal analysis, investigation, and did the study with the drug U18666A.
I Palomares-Pérez: investigation, methodology, and prepared neuronal primary cultures.
I López del Castillo: investigation and analyzed the miR-33 levels.
V Miguel: data curation and formal analysis.
L Goedeke: formal analysis, investigation, and analyzed the NPC1 expression in the brain of miR-33 KO mice.
MG Martín: investigation and made the initial discovery that brain NPC1 levels decrease with age.
S Lamas: formal analysis, provided critical analysis of the experiments and the results, revised the manuscript, and provided intellectual inputs.
H Peinado: formal analysis and facilitated the NTA analysis of EVs and revised the manuscript and provided intellectual inputs.
C Fernández-Hernando: provided the miR-33 KO mice and revised the manuscript and provided intellectual inputs.
CG Dotti: conceptualization, formal analysis, supervision, funding acquisition, designed the overall approach, coordinated the study, and drafted the manuscript and prepared the manuscript, and writing—original draft, review, and editing.

### Conflict of Interest Statement

The authors declare that the research was conducted in the absence of any commercial or financial relationships that could be construed as a potential conflict of interest.

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
