## [Reviewer comments · Life Science Alliance]

Life Science Alliance

Increased exosome secretion in neurons aging in vitro by NPC1-mediated endosomal cholesterol buildup

Francesc Guix, Ana Marrero Capitán, Álvaro Casadomé-Perales, Irene Palomares-Pérez, Inés López del Castillo, Verónica Miguel, Leigh Goedeke, Mauricio G. Martín, Santiago Lamas, Hector Peinado, Carlos Fernandez-Hernando, and Carlos Dotti

DOI: <https://doi.org/10.26508/lsa.202101055>

Corresponding author(s): Carlos Dotti, Centro de Biología Molecular Severo Ochoa and Francesc Guix, Centro de Biología Molecular Severo Ochoa

Review Timeline:

Submission Date:	2021-02-16
Editorial Decision:	2021-03-27
Revision Received:	2021-05-24
Editorial Decision:	2021-06-07
Revision Received:	2021-06-11
Accepted:	2021-06-14

Transaction Report:

March 27, 2021

Re: Life Science Alliance manuscript #LSA-2021-01055-T

Prof. Carlos G. Dotti
Centro de Biología Molecular Severo Ochoa
Molecular Neuropathology
Nicolás Cabrera 1
Madrid 28049
SPAIN

Dear Dr. Dotti,

Thank you for submitting your manuscript entitled "Increased exosome secretion in aging neurons by NPC1-mediated endosomal cholesterol accumulation" to Life Science Alliance. The manuscript was assessed by expert reviewers, whose comments are appended to this letter.

We apologize for this extended and unusual delay in getting back to you. As you will note from the reviewers' comments below, the reviewers are interested in these findings but have raised a number of significant issues, all of which should be addressed in the revised manuscript.

A common concern raised by both Rev 1 and Rev 2 is that 3 weeks in culture can not be defined as "ageing", and these concerns must be addressed in the revised manuscript for further consideration at LSA. If no additional biological data is included with regards to this concern, then the authors must explicitly state what the manuscript means by "ageing" at the beginning of the manuscript, and in title and abstract. Without any in vivo data, the title and the whole manuscript text would need to be carefully re-worded.

Thank you for this interesting contribution to Life Science Alliance. We are looking forward to receiving your revised manuscript.

Sincerely,

Shachi Bhatt, Ph.D.

Executive Editor

Life Science Alliance

<https://www.lsjournal.org/>

Interested in an editorial career? EMBO Solutions is hiring a Scientific Editor to join the international Life Science Alliance team. Find out more here -

https://www.embo.org/documents/jobs/Vacancy_Notice_Scientific_editor_LSA.pdf

B. MANUSCRIPT ORGANIZATION AND FORMATTING:

Reviewer #1 (Comments to the Authors (Required)):

In this manuscript, the authors reported increased exosome secretion in aging neurons by NPC1-mediated endosomal cholesterol accumulation, and studied the regulatory mechanism of the decline in NPC1 expression in senescent cells. The results of this study can help us understand how aging neurons maintain protein homeostasis. However, some key experiments are still lacking in experimental data, and the current data are not enough to support the author's conclusion.

1. The author needs to determine the degree of cell senescence through experiments, rather than just relying on the length of the culture time.
2. The author stated in the article that neurons cultured for more than 3 weeks were old but not dead, but there is a lack of experimental evidence that the cells are still alive.
3. Similarly, the authors have no data to prove that the degradation of proteins in aging neurons is impeded.
4. I am curious whether expressing NPC1 will cause the aging or death of neurons to accelerate?
5. The study of miR-33 is too preliminary, and more experimental results are needed to support the conclusion.

Reviewer #2 (Comments to the Authors (Required)):

The manuscript "Increased exosome secretion in aging neurons by NPC1-mediated endosomal cholesterol accumulation" by Guix et al., provides information that links NPC1 and exocytic vesicle release. In addition, the authors describe how the mTOR pathway modulates NPC1. The manuscript does need some clarifications (described below) as well as some minor experimental additions.

Proteostasis defects result in the accumulation of misfolded and/or aggregated proteins in lysosomes and phagosomes.

Notably, the authors do not demonstrate aggregated proteins in organelles.

Important clarification/concept

1. The concept of "aging" as currently used in the manuscript is not accurate. I do not think it is accurate nor consistent with literature to indicate that neurons 14-21 DIV is representative of aging. A more appropriate terminology could be maturation with a specific definite time provided by the authors in the text. This occurs throughout the text and should be altered. For example, on page 5, the authors indicate the following "neurons in culture in which the effects of age are observed" but no evidence of an aging phenotype are provided. This is especially important when considering that "neurons are from embryonic day 18 (E18) Wistar rats". Thus, even after 3 weeks in culture they can not be considered as undergone an aging process.
2. On page 6, the authors indicate that "Our cultures consist of always plating the same number of cells". However, they should quantify and state the number of neurons at time of the experiment. If neurons are lost at different rates, say for example the earlier plated neurons had greater loss than vesicles, comparison will not be as valuable.
3. Also on page 6, the text indicates the following "suggesting that the increase we see with age in vitro is genuine, due to greater release by neurons". It would be very valuable if vesicle release could be observed and photographed or video provided for direct evidence of neurons releasing vesicles.

4. On page 7, it indicates that the "vast majority of vesicles fall in the range of 50-150 nm". Please provide the exact number for the text.
5. On page 7, it states the following: "indicating that although the increased secretion of vesicles can help to detoxify the "producing" cells". Please provide data supporting the detoxifying of cells". Alternatively, cite supporting literature and make it clear this is speculation. What is the evidence that vesicles are de-toxifying neurons? Are there specific, well documented proteins that are toxic present in vesicles?
6. On page 7, the following is not accurate: "aging cortical neurons in culture closely resemble those found in cells of patients, and animal models, with Niemann Pick type C (NPC)". Note that neurons in NPC have an enlarged axon hillock (meganeurite) and the soma contains both Cholesterol and GM2 ganglioside. NPC is a lysosomal storage disorder. Therefore, this is not similar to what occurs in aging neurons.
7. On page 8, please clarify the following "In this condition, the proteostasis phenotype" by providing specific details on the phenotype.
8. On page 8, the following sentence should be modified by adding in the text in bold "resulting in the abnormal accumulation of cholesterol and GM2 ganglioside in late endocytic/MVBs and lysosomes (Zervas et al, 2001). Note that the primary pathogenic agent in NPC is GM2.
9. I believe a more appropriate disease comparator is to use Wolman's disease, which has cholesterol accumulation, but not ganglioside accumulation.
10. On page 8, the following should be changed "NPC1 is significantly downregulated in the aging in culture neurons" to "NPC1 is significantly downregulated in maturing neurons in culture.
11. Page 9. The section titled "Age-associated NPC1 downregulation is initiated by an Akt/mTORC1degradation mechanism" should have Age changed to a different term: perhaps Maturation-associated
12. On page 9. It would be useful to see if NPC1 and mTOR modulation are reciprocal: Inhibit NPC1 and ascertain if mTOR pathway is modulated. And It would be useful to inhibit NPC2 and show the specificity of effect. This is not required but it would strengthen the manuscript.
13. On page 11 (Discussion). I think a summary schematics of interactions is necessary to pull the manuscript together. Also, in the first sentence of the discussion I would strike the following "downregulation of the cholesterol transport protein NPC1 may operate as double-faced, deceiving mechanism". It is stylistic, so not necessary, but strikes me as a lot of literary license.
14. On page 12. The following needs to be amended "accumulation of cholesterol in organelles of the endo-lysosomal pathway (Zervas et al, 2001)." The cited manuscript and others from the Walkley lab actually highlight the importance of GM2 as the pathogenic agent in NPC and other storage disorders.
15. General comment on the Discussion. This section is too speculative and wandering. The discussion section would benefit greatly by being more focused and precise.

16. On Page 33 (Figure 1 legend):

1. Not sure if 14DIV and 21DIV together are reflective of "aging"
2. Aging should be defined/better yet changed to maturing
3. Note: do these findings mean that any publications showing 14DIV neurons are compromised by "aging"?
4. What type of neurons are generated from the E18 rat? Excitatory, inhibitory, interneurons. This information should be provided and shown in Supplement.

17. On page 36 (figure 2). The following modifications should be made:

- A. In panel a add a more granulated scale and truncate at 1000nm
- B. In panels c,d clarify which vesicles are quantified.

18. Figure 4a. Show XZ and YZ planes. Please provide images with Hoechst nuclear stain. Also it appears that the 24DIV cultures are not particularly healthy as there are numerous large vacuoles apparent in the images. In Figure 4C show individual measurements (dots) like in panel b

19. On page 41. Need to include the following immunolabeling pS6240 and/or pS235 from (Cell Signaling):

1. phosphorylated S6 ribosomal protein at Ser240/244 (rabbit, 1:800)
2. phosphorylated S6 ribosomal protein at Ser235/236 (rabbit, 1:100)

20. Related to Figure 8. What happens to vesicle release and is there any impact on mice?

Minor:

1. On page 5: *in vivo* should be italicized (*in vivo*)

2. On page 7: Add the following bold text...The degradation defects, lysosome enlargement (Fig. 1B,C), and increased exosome production (Fig. 2).

3. On page 35, figure legend for panel "2E". It states that "The graph shows the mean {plus minus} SEM" should be changed to :The graph shows the mean number of vesicles {plus minus} SEM

4. On page 37 (figure 4 legend. Describe what an early vs late endosome is. As it would appear to be related to 14DIV vs 21DIV, but this is not the case.... Provide clarification.

5. On page 40 (Figure 6 legend). Be direct and state the finding with figure legend: NPC1 over-expression reduces ILVs...

Also, on Page 40 (Fig. 7 legend). Change aging to "Maturing" and in panel A legend change "aging" to "time" points. (Ie Western blot analysis of total lysates of neuronal cultures at different time points).

6. Figure 7 panel a. Change the nomenclature from NPC1+ to NPCoverexpress. The + typically indicates genetics (WT).

Reviewer #3 (Comments to the Authors (Required)):

This important study demonstrates changes that occur in neurons with age. The main finding is

that an-age related decrease in proteolysis that results in accumulation of cholesterol in dysfunctional endosomal pathway is compensated by enhanced release of exosomes into the extracellular space. While such a compensatory mechanism was previously demonstrated in the brain due to neurodegenerative processes, this detailed in vitro study suggests that this type of compensation occurs during 'normal' aging. It is shown that ageing neurons in culture have a larger number of ILVs within MVBs and they secrete more small EVs than younger neurons. In an extensive study of the mechanism that leads to these changes, it is shown that the high number of ILVs is the consequence of the accumulation of cholesterol in MVBs, which in turn is due to decreased levels of the cholesterol extruding protein NPC1.

One detail that needs to be taken care of is that the small structures shown in the two left panels of Fig. 2e do not have the morphology of EVs. These are most likely lipoproteins and should not be counted as small EVs. To rule out this possibility, electron microscopy should be conducted on the supernatant and pellet of samples after ultracentrifugation at 100,000xg . While lipoproteins will be in the supernatant, EVs will be seen in the pellet. This control should be included in the manuscript. In addition, to confirm the higher level of secretion of exosomes, quantification of Western blots with antibodies to exosomal markers should be presented. A single blot, as presented in Fig. 2f and 5d is not sufficient.

Reviewer #1 (Comments to the Authors (Required)):

In this manuscript, the authors reported Increased exosome secretion in aging neurons by NPC1-mediated endosomal cholesterol accumulation, and studied the regulatory mechanism of the decline in NPC1 expression in senescent cells. The results of this study can help us understand how aging neurons maintain protein homeostasis. However, some key experiments are still lacking in experimental data, and the current data are not enough to support the author's conclusion.

1. The author needs to determine the degree of cell senescence through experiments, rather than just relying on the length of the culture time.

Answer: The experimental data showing that chronological aging *in vitro* is paralleled by the development of morphological and functional signs of maturation and aging have been published on numerous occasions, by us and many other groups, which is why we have here added a paragraph where these facts are mentioned and the original works referenced (see underlined text Page 5-6).

I include below references of some of our works in which we have shown that the chronological aging of neurons in culture is accompanied by the gradual development of biological signs of aging (understanding aging as the progressive (time-dependent) decline of biological functions).

In Martin et al., (2008), (*Mol Biol Cell*. 2008 May;19(5):2101-12. doi: 10.1091/mbc.e07-09-0897) and Martin et al. (2014) (*EMBO Mol Med*. 2014 Jul;6(7):902-17. doi: 10.15252/emmm.201303711) we demonstrated that neuronal aging *in vitro* is accompanied by changes in membrane lipid composition and fluidity resulting in the re-organization -and level of activity- of survival and plasticity receptors

In Trovò et al. (2014) (*Nature Neurosci*. 2013 Apr;16(4):449-55. doi: 10.1038/nn.3342) we demonstrate that neuronal aging *in vitro* induces the downregulation of PI(4,5)P2 resulting in reduced activity of the molecular machinery involved in long term potentiation (LTP).

In Palomer et al (2016) (*Cell Reports* 2016 Sep 13;16(11):2889-2900. doi: 10.1016/j.celrep.2016.08.028) we showed that neurons aging in vitro present a repressed gene expression pattern for a synaptic plasticity gene (BDNF) due to the activation of epigenetic repressive mechanisms, contributing to reduced learning and memory mechanisms.

In Benvegnù et al (2017) (*Molecular Cell*. 2017 May 4;66(3):358-372.e7. doi: 10.1016/j.molcel.2017.04.005) we showed that neuronal aging *in vitro* is accompanied by a reduction in proteosomal activity resulting in monoubiquitination of a E3 ubiquitin ligase and its shuffling to the nucleus where it contributes to improve survival under stress.

In Martín-Segura et al (2019) (*Aging Cell*. 2019 Jun;18(3):e12932. doi: 10.1111/acel.12932) we showed that neurons in culture develop insulin resistance due

to receptor chronic activity and pathway desensitization, similarly to the *in situ* occurrence of brain insulin resistance with age.

2. The author stated in the article that neurons cultured for more than 3 weeks were old but not dead, but there is a lack of experimental evidence that the cells are still alive.

In the new Supplementary figure S1B we present data showing the levels of lactate dehydrogenase in 3-week old neurons in culture, demonstrating that these cells are alive. In the new Supplementary Figure S1C we show that levels of protein remain unchanged with time *in vitro*. The description of these results are provided in the main text (underlined, page 7)

We regret not having been sufficiently explicit in regard to the live/death balance of neurons in culture, which we have demonstrated in some of our previous publications. As a more significant example, in *Martin et al. 2011* we quantified subG0/G1 hypodiploid nuclei in cultured neurons at different times *in vitro* by cytofluorimetry and the levels of DNA fragmentation by the TUNEL assay (*Neurobiol Aging. May;32(5):933-43. doi: 10.1016/j.neurobiolaging. 2009.04.022*).

In *Martin et al. 2014 (EMBO Mol Med. Jul;6(7):902-17)* we demonstrated that 3-week old *in vitro* neurons present proper receptor internalization and receptor lateral diffusion.

In *Benvegnù et al (Molecular Cell. 2017 May 4;66(3):358-372.e7. doi: 10.1016/j.molcel.2017.04.005)* we demonstrated cell survival activity in 3 week-old neurons *in vitro*.

3. Similarly, the authors have no data to prove that the degradation of proteins in aging neurons is impeded.

In the new Supplementary Figure S1A we now show that time (age) *in vitro* increases the levels of ubiquitinated proteins (Fig. S1A).

In the original version, reduced degradative capacity of neurons with time *in vitro* was reflected by: 1) the existence of more and larger lysosomes with age *in vitro* (Fig. 1A-C), and 2) reduced levels of LAMP 2A, a mediator of chaperone-mediated autophagy, with age *in vitro* (Fig. 1 D).

These evidences of defects in proteostasis with time (age) *in vitro* are mentioned in the corrected version (underlined text, pages 6 and 7).

In addition,

In previous works we demonstrated that neurons aging *in vitro* gradually accumulate undegraded material in the lysosomes (*Martin et al. 2011 Neurobiol Aging. May;32(5):933-43*). Furthermore, in a more recent work we demonstrated altered proteasomal degradation in neurons aging *in vitro* due to polyubiquitination of the ubiquitin ligase Mahogunin (see *Benvegnù et al., 2017 Molecular Cell. 2017*). Other

laboratories have also demonstrated that neurons aging in culture present defects in protein and membrane degradation mechanisms (Moreno-Blas D, et al (2019). *Cortical neurons develop a senescence-like phenotype promoted by dysfunctional autophagy. Aging* 30;11(16):6175-6198).

4. I am curious whether expressing NPC1 will cause the aging or death of neurons to accelerate?

In the corrected version we describe that even if few neurons could be induced to over-express NPC1, this did not result in an obvious alteration of the cells' phenotype, except for a reduced number of intraluminal vesicles. This is now clarified in the text (underlined, Page 10).

5. The study of miR-33 is too preliminary, and more experimental results are needed to support the conclusion.

We include new data (Supplementary Fig. S9) showing the levels of ABCA1, another target of miR-33, in the brain of miR-33 KO mice. We have also added to Figure 8B a quantification of the decreased levels of NPC1 (N=4 independent cultures; Figure 8C) and ABCA1 (N=3 independent cultures; Figure 8D) in cells transfected with the miR-33 mimic. This effect is also described in the text (underlined, page 13)

Reviewer #2 (Comments to the Authors (Required)):

The manuscript "Increased exosome secretion in aging neurons by NPC1-mediated endosomal cholesterol accumulation" by Guix et al., provides information that links NPC1 and exocytic vesicle release. In addition, the authors describe how the mTOR pathway modulates NPC1. The manuscript does need some clarifications (described below) as well as some minor experimental additions.

Proteostasis defects result in the accumulation of misfolded and/or aggregated proteins in lysosomes and phagosomes.

Notably, the authors do not demonstrate aggregates proteins in organelles.

In the new supplementary Figure S1A we demonstrate that age *in vitro* is accompanied by the accumulation of ubiquitinated proteins, reflecting proteasomal dysfunction.

We would like to point out that we also presented data demonstrating reduced degradative capacity of neurons with time *in vitro*: 1) Fig. 1 A-C demonstrates the existence of more and larger lysosomes with age *in vitro* (i.e. accumulation of undegraded material), and 2) Fig. 1 D shows reduced levels of LAMP 2A, a mediator of chaperone-mediated autophagy, with age *in vitro*.

We regret not having clarified in the previous version that we had previously demonstrated the aggregation of proteins in cultured neurons over time *in vitro*. Thus, as early as 2011, we demonstrated that neurons *in vitro* slowly and progressively accumulate undegraded material in the lysosomes with time *in vitro* (Martin et al. 2011 *Neurobiol Aging*. May;32(5):933-43).

Furthermore, in a more recent work we demonstrated altered degradation mediated by the proteasome due to polyubiquitination of the ubiquitin ligase Mahogunin (see Benvegnù et al., 2017 *Molecular Cell*. 2017).

Other laboratories have also demonstrated that neurons aging in culture present defects in protein and membrane degradation mechanisms (Moreno-Blas D, et al (2019). *Cortical neurons develop a senescence-like phenotype promoted by dysfunctional autophagy*. *Aging* 30;11(16):6175-6198).

Important clarification/concept

1. The concept of "aging" as currently used in the manuscript is not accurate. I do not think its accurate nor consistent with literature to indicate that neurons 14-21 DIV is representative of aging. A more appropriate terminology could be maturation with a specific definite Tian provided by the authors in the text. This occurs throughout the text and should be altered. For example, on page 5, the authors indicate the following "neurons in culture in which the effects of age are observed" but no evidence of an aging phenotype are provided. This especially important when considering that

"neurons are from embryonic day 18 (E18) Wistar rats". Thus, even after 3 weeks in culture they can not be considered as undergone an aging process.

In its strict significance the term "aging" refers to the chronological process of becoming older. In biological terms, "aging" refers to the decay of function with time. We have here compared 3-4 week-old neurons with 1-2 week-old neurons. Taking into account that these cells die, in the conditions here used, after the 4th week *in vitro*, the analysis of 3-4 week old neurons *in vitro* reflect their aging phase (compared to 1-2 week old neurons). Furthermore, the 3-4 week-old neurons in culture show reduced functionality compared with 1-2 week-old neurons. Therefore, the use of the term "aging" is relevant in the context of our experimental model but should not be compared with other models with different lifespans.

Comment to this reviewer about studying aging *in vitro*:

aging-in-a-dish is an experimental model to generate basic knowledge on aging. Data arising from this model should not be extrapolated to other models (like aging *in vivo*). On the other hand, the data arising from the *aging-in-the-dish* models may facilitate, the study and understanding of the *in vivo* situation. We apologize if in our manuscript we implied that the mechanisms that operate in cultured neurons also operate in neurons *in situ*. We have corrected in this version all those phrases that can be interpreted in this sense. On page 5 of this corrected version we make a description of the stages of maturation of embryonic neurons of the mouse brain once seeded in culture, and the implications from results based on this model (underlined text, page 5)

Regarding this comment from the reviewer:

This especially important when considering that "neurons are from embryonic day 18 (E18) Wistar rats". Thus, even after 3 weeks in culture they can not be considered as undergone an aging process.

It is difficult for me to understand this type of reasoning. Does the reviewer imply that embryonic neurons remain embryonic *in vitro*, no matter for how long they are maintained the *in vitro* condition?

Embryonic neurons (or any embryonic cell) progress through all the stages of life when seeded *in vitro*; different embryonic cells have different lifespans *in vitro*. Regarding neurons, in the few first hours after seeding they become "newborn" neurons, characterized by the sprouting of axons and dendrites. After a day or two, they start to mature into a "young" neuron phenotype, ending by the end of the first week with properly formed axons and dendrites. In the third week, the initial embryonic neurons have acquired characteristics of mature neurons, with dendritic specialization (spines, postsynaptic specializations), proper mechanisms of sorting proteins and lipids to their different destinations, clustered receptors into domains and subdomains, and electrophysiological characteristics of LTP and LTD. And then, they start to show signs of wear-and-tear (aging): proteins are not degraded properly and accumulate, also ROS accumulate, aberrant phosphorylation events take place, gene expression is altered, trophic factor signaling is impaired, membrane lipids are removed. And finally,

the neurons that one day were embryonic die, in the 4th week after taking from the embryos.

Conclusion: like any cell, embryonic neurons progress through the four basic phases of life when seeded *in vitro*, only that they do so in the course of 4 weeks. That is the lifespan of embryonic neurons after seeding *in vitro*. Is this model useful to understand aging of neurons *in vivo*? Is the study of mouse aging useful to understand human aging? These are all experimental models, some advantageous for certain aspects and others for different matters. A good comprehension will most likely come from the studies in many different models.

Note: I am sure that this reviewer has a more than interesting and elaborate reason for having written that neurons obtained from an embryo could not age in culture. For a matter of collegiality, I would very much appreciate knowing that opinion.

2. *On page 6, the authors indicate that "Our cultures consist of always plating the same number of cells". However, they should quantify and state the number of neurons at time of the experiment. If neurons are lost at different rate say for example the earlier plated neurons had greater loss then vesicles comparison will not be as valuable.*

We now include two new experiments: 1) analysis of release of lactate dehydrogenase (LDH) to the media as an indicator of cell viability, and 2) compared the total protein content, determined by BCA, in each culture where we determined the EVs concentration by Nanosight in the media. The relative amount of LDH released to the media is similar between 12-14 DIV, 15-21 DIV and 22-28 DIV cultures (Supplementary Figure S1B). The total protein content does not vary during maturation of the cultures *in vitro* (Supplementary Figure S1C). These results indicate that cell loss is not significant in this model of *in vitro* aging.

3. *Also on page 6, the text indicates the following "suggesting that the increase we see with age in vitro is genuine, due to greater release by neurons". It would be very valuable if vesicle release could be observed and photographed or video provide for direct evidence of neurons releasing vesicles.*

We are analyzing "nanovesicles". Direct visualization and quantification of **nanovesicles** by fluorescence (vesicles whose size is in the nanometer range) in cells like neurons, highly sensitives to light-induced damage, is beyond current technical possibilities. Current state-of-the-art methods for vesicle quantification are Nanosight, FACS and electron microscopy. We used these methods in the current work, therefore allowed to conclude that neurons in culture undergo a genuine sequential (time-associated) increase in vesicle secretion.

4. *On page 7, it indicates that the "vast majority of vesicles fall in the range of 50-150 nm". Please provide the exact number for the text.*

We have now added the data corresponding to number of vesicles with different sizes. See text (underlined, page 8). The text now reads:

mean percentage of vesicles \pm 95% Confidence Interval: 65.59 % \pm 4.09 % for 12-14 DIV; 71.8 % \pm 4.55 % for 15-21 DIV; 77.69 % \pm 5.03 % for 22-29 DIV

5. *On page 7, it states the following: "indicating that although the increased secretion of vesicles can help to detoxify the "producing" cells". Please provide data supporting the detoxifying of cells". Alternatively, cite supporting literature and make it clear this is speculation. What is the evidence that vesicles are de-toxifying neurons? Are there specific, well documented proteins that are toxic present in vesicles?*

On page 8 (underlined text) we describe that exosomes from 3-4 week *in vitro* neurons contain higher levels of the microtubule-associated protein tau (see Results, underlined text on page 8 and Supplementary Figure S4) compared to exosomes from 1-2 week *in vitro* neurons. References to this result and previous works supporting the detoxifying role of exosomes are now provided in Discussion (underlined text, page 14).

6. *On page 7, the following is not accurate: "aging cortical neurons in culture closely resemble those found in cells of patients, and animal models, with Niemann Pick type C (NPC)". Note that neurons in NPC have an enlarged axon hillock (meganeurite) and the soma contains both Cholesterol and GM2 ganglioside. NPC is a lysosomal storage disorder. Therefore, this is not similar to what occurs in aging neurons.*

We have changed this sentence, that now reads as follows: The degradation defects, lysosome enlargement (Fig 1B and C), and increased exosome production (Fig 2) observed in aging cortical neurons in culture reproduce some of the features found in cells of patients, and animal models, with Niemann Pick type C (NPC) disease. (underlined text, page 9)

7. *On page 8, please clarify the following "In this condition, the proteostasis phenotype" by providing specific details on the phenotype.*

This has been clarified (autophagic-lysosomal dysfunction), se underlined text page 9

8. *On page 8, the following sentence should be modified by adding in the text in bold "resulting in the abnormal accumulation of cholesterol and GM2 ganglioside in late endocytic/MVBs and lysosomes (Zervas et al, 2001). Note that the primary pathogenic agent in NPC is GM2.*

This has been corrected (underlined text page 9).

9. *I believe a more appropriate disease comparator is to use Wolman's disease, which has cholesterol accumulation, but not ganglioside accumulation.*

We considered that the phenotype observed in *in vitro* aged neurons resembled Niemann Pick Disease type C phenotype due to: a) the neuronal component of NPC disease and b) the occurrence in NPC1 of features also observed in our *in vitro* model namely accumulation of intracellular vesicles and higher release of exosomes.

10. On page 8, the following should be changed "NPC1 is significantly downregulated in the aging in culture neurons" to "NPC1 is significantly downregulated in maturing neurons in culture."

We have already explained the meaning of "aging" and added the term "*in vitro*" to the title of this manuscript so readers understand from the very beginning that the results are from an aging-in-the-dish model. No implications are made on relevance of our results to *in vivo* aging, mammalian aging, or human aging. Time will tell to which extent the basic knowledge generated here applies to more complex scenarios.

11. Page 9. The section titled "Age-associated NPC1 downregulation is initiated by an Akt/mTORC1 degradation mechanism" should have Age changed to a different term: perhaps Maturation-associated

We have changed this sub-title to **NPC1 downregulation in cortical neurons aging in culture is initiated by an Akt/mTORC1- degradation mechanism** (underlined text, page 11).

12. On page 9. It would be useful to see if NPC1 and mTOR modulation are reciprocal:

We have addressed this query experimentally. The new data are shown in the new Supplementary Figure (Fig S8) and the result explained in Results (Underlined text, page 12).

As suggested by the reviewer, we inhibited NPC1 in 14 DIV neurons and checked the levels of the phosphorylated form of the mTOR substrate S6K. The inhibition of NPC1 decreased the levels of the phosphorylated form of S6K. Decreased mTOR activity resulting from impaired NPC1 function may be a form of negative feedback intended to prevent excessive mTOR-induced NPC1 degradation. However, this potential regulatory mechanism, in our *in vitro* system, can be by-passed by the up-regulation of miR-33, what prevents the restoration of the NPC1 levels.

13. On page 11 (Discussion). I think a summary schematics of interactions is necessary to pull the manuscript together. Also, in the first sentence of the discussion I would strike the following "downregulation of the cholesterol transport protein NPC1 may operate as double-faced, deceiving mechanism". It is stylistic, so not necessary, but strikes me as a lot of literary license.

We have changed the first sentence of Discussion (less literary, more "scientific"). We have also added a schematics of the possible interactions suggested by our results (new Figure 9). The legend for Figure 9 can be found in page 42.

14. On page 12. The following needs to be amended "accumulation of cholesterol in organelles of the endo-lysosomal pathway (Zervas et al, 2001)." The cited manuscript and others from the Walkley lab actually highlight the importance of GM2 as the pathogenic agent in NPC and other storage disorders.

This has been changed according to suggestion (see underlined text, page 9)

15. General comment on the Discussion. This section is too speculative and wandering. The discussion section would benefit greatly by being more focused and precise.

We have partly rewritten the Discussion, and to our knowledge it is now better focused.

16. On Page 33 (Figure 1 legend):

1. Not sure if 14DIV and 21DIV together are reflective of "aging"

Neurons of 21 days *in vitro* are, chronologically and biologically speaking, one week older than 14 days *in vitro* and a few days from dying, therefore chronologically "old" in this particular system. Furthermore, they are also biologically old: their lysosomes are larger, they secrete more exosomes, the proteasome works less efficiently, they accumulate phosphorylated tau, trophic factors signal poorly, they show it occurs receptor signaling desensitization, learning and memory genes are poorly expressed, membrane lipids change, lipofuscin accumulates in lysosomes (this and previous works).

2. Aging should be defined/better yet changed to maturing

We have already stated the meaning of cellular "aging": the biological changes occurring with time closest to the time of cells' death.

3. Note: do these findings mean that any publications showing 14 DIV neurons are compromised by "aging"?

In an *in vitro* system aging of the neurons (or of any cell) must be linked to the lifespan of the cells. Thus, 14 days *in vitro* neurons could be considered "aged" if their lifespan in culture is 16-20 days. In our system, cultured neurons begin to die when they are 25-28 days *in vitro*, reason by which we consider, aging those neurons closest to the death process, that is, those maintained *in vitro* for 21 or more days

4. What type of neurons are generated from the E18 rat? Excitatory, inhibitory, interneurons. This information should be provided and shown in Supplement.

Cultures of dissociated cortical neurons from rat embryos have been used since pioneering works from the 70's (Dichter MA. Rat cortical neurons in cell culture: culture methods, cell morphology, electrophysiology, and synapse formation. Brain

Res. 1978 Jun 30;149(2):279-93). The early works defined that these cortical neurons in culture highly resembled cortical neurons *in situ* both in terms of morphology (pyramidal neurons) and in the type of synaptic contacts they made. This is now clarified in the text (underlined, page 18).

**17. On page 36 (figure 2). The following modifications should be made:
In panel a add a more granulated scale and truncate at 1000nm**

Panel A of Figure 2 has been changed to show a more granulated scale and truncated at 1000 nm.

B. In panels c,d clarify which vesicles are quantified.

In both Panel C and D, the data used to build the plots come from Nanosight quantifications performed in the supernatant obtained after centrifuging the neuronal media at 10,000 g to remove most of the microvesicles and larger vesicles (the supernatant is diluted 1 in 10 before applying the sample to the Nanosight device). In both cases (Panel C and D), all vesicles present in this supernatant are quantified. Panel D shows the vesicles produced in a 24h period, and the concentration of vesicles is determined from the area under the curve of the Nanosight profile shown in panel B and multiplied by 10 (dilution factor of the sample). Panel C shows the vesicles present in the media accumulated over time, and the concentration of vesicles is determined from the area under the curve of Nanosight profiles similar to the ones shown in Panel B.

18. Figure 4a. Show XZ and YZ planes. Please provide images with Hoechst nuclear stain. Also it appears that the 24DIV cultures are not particularly healthy as there are numerous large vacuoles apparent in the images. In Figure 4C show individual measurements (dots) like in panel b

The new Figure 4A shows cells with the DAPI staining to visualize the nucleus. In addition, two new Supplementary Figures have been added (Figs S5 and S6), which show the co-localization of Bodipy-cholesterol and the early endosomal marker EEA1 (Fig S5) or the late endosomal marker LIMP-I (Fig S6) in the X-Z and the Y-Z planes. The legends for the added supplementary figures can be found in page number 45.

19. On page 41. Need to include the following immunolabeling pS6240 and/or pS235 from (Cell Signaling):

1. phosphorylated S6 ribosomal protein at Ser240/244 (rabbit, 1:800)
2. phosphorylated S6 ribosomal protein at Ser235/236 (rabbit, 1:100)

We did not follow this suggestion, we consider that the data presented regarding the involvement of the Akt/mTOR pathway sufficiently prove how aging *in vitro* could affect NPC1 levels. This is properly discussed in the current manuscript.

20. *Related to Figure 8. What happens to vesicle release and is there any impact on mice?*

Not sure we understand this question: Does the reviewer mean what is the effect on synaptic vesicle release? This is certainly a most interesting question for which we have no answer at the moment.

Minor:

1. *On page 5: in vivo should be italicized (in vivo)*

We have searched and changed in vivo for *in vivo*

2. *On page 7: Add the following bold text...The degradation defects, lysosome enlargement (Fig. 1B,C), and increased exosome production (Fig. 2).*

We have made the suggested change

3. *On page 35, figure legend for panel "2E". It states that "The graph shows the mean {plus minus} SEM" should be changed to: The graph shows the mean number of vesicles {plus minus} SEM*

We changed this sentence.

4. *On page 37 (figure 4 legend. Describe what an early vs late endosome is. As it would appear to be related to 14DIV vs 21DIV, but this is not the case.... Provide clarification.*

The term early or late endosomes is used to distinguish endosomes by the time it takes for endocytosed material to reach them. Early endosomal compartments are the first stage for internalized material and can be specifically identified with the markers EEA1 or Rab5. Late endosomal compartments (including multivesicular bodies) are the consequence of the maturation of early endosomes and can be identified with the markers LIMP-I or Rab7. A clarification for these terms has been added to the legend of Figure 4 (text underlined, page 35).

5. *On page 40 (Figure 6 legend). Be direct and state the finding with figure legend: NPC1 over-expression reduces ILVs... Also, on Page 40 (Fig. 7 legend). Change aging to "Maturing" and in panel A legend change "aging" to "time" points. (Ie Western blot analysis of total lysates of neuronal cultures at different time points).*

We changed the figure legend of Figure 6 by "NPC1 over-expression reduces ILVs in aged neurons".

Regarding the term “Aging”, and as stated above, our position is that these cells suffer the passage of time and are close to the end of their lifespan, therefore in the “aging” phase (in culture).

6. Figure 7 panel a. Change the nomenclature from NPC1+ to NPCoverexpress. The + typically indicates genetics (WT).

In Figure7, we switched “NPC1+” by “NPC overexpression”

Reviewer #3 (Comments to the Authors (Required)):

This important study demonstrates changes that occur in neurons with age. The main finding is that an-age related decrease in proteolysis that results in accumulation of cholesterol in dysfunctional endosomal pathway is compensated by enhanced release of exosomes into the extracellular space. While such a compensatory mechanism was previously demonstrated in the brain due to neurodegenerative processes, this detailed in vitro study suggests that this type of compensation occurs during 'normal' aging. It is shown that ageing neurons in culture have a larger number of ILVs within MVBs and they secrete more small EVs than younger neurons. In an extensive study of the mechanism that leads to these changes, it is shown that the high number of ILVs is the consequence of the accumulation of cholesterol in MVBs, which in turn is due to decreased levels of the cholesterol extruding protein NPC1. One detail that needs to be taken care of is that the small structures shown in the two left panels of Fig. 2e do not have the morphology of EVs. These are most likely lipoproteins and should not be counted as small EVs. To rule out this possibility, electron microscopy should be conducted on the supernatant and pellet of samples after ultracentrifugation at 100,000xg. While lipoproteins will be in the supernatant, EVs will be seen in the pellet. This control should be included in the manuscript.

We carried out the electron microscopy analysis of the supernatant and the pellet at 100,000g as suggested by this reviewer (data shown in Supplementary Fig. S3). This experiment revealed that the vast majority of structures with the appearance of EVs appear in the pellet.

In addition, to confirm the higher level of secretion of exosomes, quantification of Western blots with antibodies to exosomal markers should be presented. A single blot, as presented in Fig. 2f and 5d is not sufficient.

Following this reviewer's indication, the new Supplementary Figure S4 is the representation of 2 new experiments to quantify HSP90, CD81 and Tau in exosomes isolated from the media of 14 and 21 DIV neuronal cultures.

The new Supplementary Figure S7 shows the results of two new experiments to determine CD81 content in exosomes isolated from control or U18666A-treated neuronal cultures.

June 7, 2021

RE: Life Science Alliance Manuscript #LSA-2021-01055-TR

Prof. Carlos G. Dotti
Centro de Biología Molecular Severo Ochoa
Molecular Neuropathology
Nicolás Cabrera 1
Madrid 28049
Spain

Dear Dr. Dotti,

Thank you for submitting your revised manuscript entitled "Increased exosome secretion in neurons aging in vitro by NPC1-mediated endosomal cholesterol buildup". We would be happy to publish your paper in Life Science Alliance pending final revisions necessary to meet our formatting guidelines.

- please add ORCID ID for secondary corresponding author-they should have received instructions on how to do so
- please rename the "Competing Interest" section to "Conflict of Interest statement"
- please add your main, supplementary figure, and table legends to the main manuscript text after the references section
- please add callouts for Figures 1C; 3A, B, D; 6A-C; S3A-D; S4A-D; S8A, B, and S9A, B to your main manuscript text

A. FINAL FILES:

-- High-resolution figure, supplementary figure and video files uploaded as individual files: See our detailed guidelines for preparing your production-ready images, <https://www.life-science->

alliance.org/authors

B. MANUSCRIPT ORGANIZATION AND FORMATTING:

Sincerely,

Reviewer #1 (Comments to the Authors (Required)):

in this version the authors answered all my questions, I think the conclusions of the paper justified based on the presented data.

Reviewer #3 (Comments to the Authors (Required)):

All the reviewers' comments were responded to in detail.

June 14, 2021

RE: Life Science Alliance Manuscript #LSA-2021-01055-TRR

Prof. Carlos G. Dotti
Centro de Biología Molecular Severo Ochoa
Molecular Neuropathology
Nicolás Cabrera 1
Madrid 28049
Spain

Dear Dr. Dotti,

Thank you for submitting your Research Article entitled "Increased exosome secretion in neurons aging in vitro by NPC1-mediated endosomal cholesterol buildup". It is a pleasure to let you know that your manuscript is now accepted for publication in Life Science Alliance. Congratulations on this interesting work.

DISTRIBUTION OF MATERIALS:

Again, congratulations on a very nice paper. I hope you found the review process to be constructive and are pleased with how the manuscript was handled editorially. We look forward to future exciting submissions from your lab.

Sincerely,
